# COPER: Agentic Context Improves and Stabilizes LLMs in Multi-Player Game

## Abstract

Recent multi-player game benchmarks can be sensitive: modest changes to role, system, or judge prompts often flip win-rate rankings under identical decoding; and static, read-once descriptions fail to impart the *game-specific priors* (rules, legality, action→transition effects) needed for consistent play. We document this *context-induced* instability and argue evaluation should be *agentic*: let interaction surface and solidify priors, then evaluate models for both their strength (performance) and reliability (consistency under perturbations). To establish more reliable baselines, we present **COPER**, a backbone-agnostic, tuning-free self-play recipe that (i) *evolves prompts* using a conservative TrueSkill lower-confidence bound, (ii) writes structured reflections into a persistent *experience bank* retrieved across turns to supply rule-aware priors, and (iii) uses *prioritized replay* to revisit rare, informative states for sample-efficient stabilization. Across five text games, COPER raises mean win rate from **24.9%** → **49.5%** (GPT-4o-mini) and **21.7%** → **44.3%** (Qwen-2.5-7B-Instruct) with a small budget (5×400 self-play games per task), and *stabilizes* agent performance under evaluation. These results show that much of today's LLM game headroom can be unlocked by *context* rather than weight updates, with COPER yielding strong improvements in negotiation games, competitive results in some imperfect-information settings, and RL remaining more effective in perfect-information games.

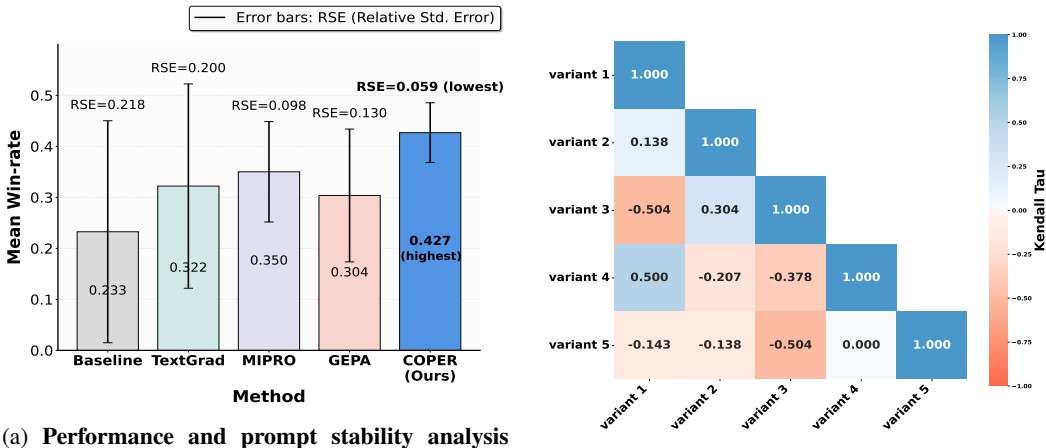

(a) **Performance and prompt stability analysis across different prompt optimization methods.**

(b) **Ranking sensitivity in KUHNPOKER.**

Figure 1: **Left:** We evaluate baseline and other prompt optimization methods by average win-rate and RSE. Our method, COPER, achieved the highest win-rates and the lowest variance, demonstrating enhanced performance and stability. **Right:** With environment and evaluator pools fixed, five *nearly equivalent* prompt variants still flip pairwise outcomes and reshuffle rankings (detailed prompt variants are presented in Appendix D.4). The heatmap shows Kendall's $\tau_b$ for every pair of prompts: blue means very similar rankings ($\tau_b \approx 1$), white means unstable rankings ($\tau_b \approx 0$), and orange means rank reversals ($\tau_b < 0$).

# 1 INTRODUCTION

Large language models (LLMs) have rapidly saturated many static benchmarks, leaving limited headroom for further progress on single-turn QA and reasoning datasets such as AIME (AIME, 2024), SWE-Bench (Jimenez et al., 2023), and GPQA (Rein et al., 2024). This saturation has shifted attention toward multi-step evaluations, especially *game-based* benchmarks (Yao et al., 2025; Duan et al., 2024; Topsakal et al., 2024; Fan et al., 2024), which stress long-horizon reasoning and adaptation. Games are a natural testbed: they are easy to simulate, come with well-defined win conditions, and demand capabilities that mirror real-world challenges, e.g., planning under uncertainty, coordination, negotiation, and context adaptation.

Unfortunately, current LLM evaluations are found to be sensitive and under-agentic.Razavi et al. (2025) Firstly, current game evaluations are *prompt sensitive*. Prior work has shown that LLM accuracy can be highly sensitive to prompt phrasing in QA (Mizrahi et al., 2024), and this effect is amplified in interactive games where agents exchange information over many turns. Because prompts couple across agents, judges, and tools, small changes in role or system templates can flip ELO comparisons and reorder models under identical decoding (Fig. 1b). Our measurements reproduce this phenomenon: near equivalent prompts induce large variations in win rates and produce ranking reversals. As shown in Fig. 1b, in KUHNPOKER, holding the evaluation model fixed, even minor wording changes in the initial game prompt led to ranking reversals (orange cells), as measured by Kendall's $\tau_b$ between leaderboards.

Secondly, read-once descriptions lack the *agentic* feedback loop necessary to develop *game-specific priors* such as precise rules, legality constraints, and the effects of actions on game states and payoffs. While benchmarks Yao et al. (2025); Guertler et al. (2025a) provide textual rule descriptions, these priors are rarely internalized from a single reading. Without interaction-driven learning, models repeatedly violate rules and exhibit poor long-term strategic playYao et al. (2025). Unlike Olympiad-style problems that can be solved through careful reasoning alone, games require continuous interaction. This is where players must refine their understanding of the game mechanics and adapt their strategies dynamically based on experience.

**Thesis** We argue that game-LLM evaluation should *mirror human play*: let interaction surface and solidify priors, then evaluate models for both strength and reliability. We therefore seek a weight tuning-free, agentic evaluation recipe that stabilizes rankings under prompt variation and closes the gap between a model's latent competence and realized in-game performance.

**Approach** We propose **COPER** (Context Optimized with Prompt, Experience and Replay), an LLM tuning-free framework that pairs (i) **prompt evolution**, (ii) a persistent **experience bank**, and (iii) **prioritized replay**. Prompt evolution treats prompt selection as structured search with a conservative TrueSkill lower-confidence bound $S(p) = \mu - \kappa\sigma$ (Eq.1) to favor candidates that are strong and *reliable*. The experience bank consolidates trajectory reflections using CRUD-style updates (Eq. 3) and retrieves relevant insights to refresh the operative description across turns, surfacing rule-aware priors without weight updates. Replay mixes fresh self-play with targeted revisits to rare/informative states via a lightweight gate and priority exponent, accelerating stabilization while preserving coverage (see Eq. 4 and Eq. 5). Together, these components create an *agentic context* at inference time—*prompt + experience + replay*—that improves adherence to rules and reduces variance across prompts.

Across five text-based games sampled from `SPIN-Bench` (Yao et al., 2025) and `TextArena` (Guertler et al., 2025a), COPER achieves large, budget-efficient gains and more reliable rankings under prompt stratification: for **GPT-4o-mini** (OpenAI, 2024), mean win rate improves from **24.9%** to **49.5%**; for **Qwen-2.5-7B-Instruct** (Yang et al., 2024), from **21.7%** to **44.3%**, using only $5 \times 400$ self-play games per task. Moreover, rankings stabilize when the evaluation protocol itself is agentic.

Our contributions can be outlined as follows:

- **Context-induced ranking instability in multi-agent LLM game benches**. Auditing `SPIN-Bench` and `TextArena` showed that baselines are highly sensitive to how agents, judges and tools are prompted; because prompts couple across agents, small changes in role/system prompts or message templates can flip ELO and model orderings even with identical de-

coding settings. This effect, amplified by cross-agent interactions and path-dependent dialog, goes beyond single-LM prompt sensitivity and is under-reported despite benches exposing rich prompt hooks. We argue that multi-prompt, prompt-stratified reporting should be mandatory for these benches.

- **Prompt + Experience + Replay (COPER): a simple yet effective recipe.** We introduce a weight-tuning-free mechanism that (i) writes episodic summaries of trajectories to an experience bank, (ii) retrieves them to edit/evolve per-agent prompts online, and (iii) performs branch-and-replay from flagged states during self-play to guide exploration - thus "fine-tuning without weight updates." COPER unifies verbal-feedback reflection with memory-augmented prompting and experience replay into a single, principled procedure tailored to multi-agent games.

- **Large, sample-efficient gains on multi-LLM games.** As shown in Fig. 1a, COPER achieves substantially higher win rates than prior prompt optimization methods like `MIPRO`, while staying competitive with RL-based baselines like `UnstableBaseline` at a fraction of their computational and rollout cost.

## 2 PRELIMINARY AND PROBLEM STATEMENT

**Two-Player Multi-Turn Markov Game.** We consider a two-player, turn-based, zero-sum, partially observable environment defined by the tuple $(S, A, O, T, R, \Omega)$. Here, $S$ denotes the state space, $A$ the action space, $O$ the observation space, $T : S \times A \to S$ the transition kernel, and $R : S \times A \to \{-1, 0, 1\}$ is a sparse terminal reward. In general, the agent does not observe the full state $s \in S$; instead, it receives a partial observation $o = \Omega(s)$, where $\Omega : S \to O$ is the observation function mapping states to agent observations. Players alternate turns, and we let $p \in \{0, 1\}$ denote the player index. At time step $t$, the active player $p = (t \bmod 2)$ selects an action $a_t^{(p)} \in A$, while the opponent remains idle. Terminal outcomes are given by $R_0(\tau) = \rho(s_T)$ and $R_1(\tau) = -\rho(s_T)$, where $\tau$ denotes the trajectory of play and $\rho : S^{\text{terminal}} \to \{-1, 0, 1\}$ assigns each terminal state $s_T$ to a final outcome. As the interaction length increases, sampling noise, non-stationarity, and error propagation accumulate, resulting in amplified variance in the observed outcomes.

**Game Context: Prompt and Experience.** We use *context* to denote all information that conditions the model before and during play. Let $C = (q, M)$, where $q$ is the **instruction prompt**: role and core system text fixed at the start of play; $M$ is the **experience memory**, interaction-derived knowledge distilled from self-play and evaluation trajectories and retrieved at inference without weight updates. This contextual prior helps the model interpret transitions and payoffs more effectively, promoting stability over extended interactions.

**Full-Context Evaluation.** Given a method $m$ with context design space $\mathcal{C}_m$ (e.g., choices of $p$ and $M$), game suite $\mathcal{G}$, and opponent pool $\mathcal{E}$, each independent run produces a best context $C_r \in \mathcal{C}_m$. We execute $n$ runs. For every game $g \in \mathcal{G}$ and opponent $e \in \mathcal{E}$, we play $k$ rounds; each round consists of two games with swapped first-move order to remove first-move bias. Opponent models use fixed reference contexts specified in Appx. D. Let $\text{WR}_{r,g} \in [0, 1]$ denote the evaluated agent's win rate in game $g$ under context $C_r$, averaged over all opponents in $\mathcal{E}$ and $k$ rounds. The overall performance for run $r$ is then $x_r = \frac{1}{|\mathcal{G}|} \sum_{g \in \mathcal{G}} \text{WR}_{r,g}$. We report mean performance across runs, $\text{mean}(x_1, \ldots, x_n)$, together with *relative standard error* (RSE), defined as $\text{RSE}(\%) = 100 \times \frac{\text{std}(x_1, \ldots, x_n)}{\text{mean}(x_1, \ldots, x_n) \times \sqrt{n}}$, where lower RSE values indicate greater stability across independent context selections.

**Variance Across Prompt and Context.** Small wording changes in the prompt template can induce large shifts in both absolute and relative performance, which motivates *multi prompt* evaluation and calibration protocols (Mizrahi et al., 2024; Zhao et al., 2021). We evaluate *state-of-the-art* models (GPT-4o (OpenAI et al., 2024), DeepSeek-R1 Guo et al. (2025), Gemini-2.5-Flash Comanici et al. (2025), Grok-3-Mini (xAI, 2025), GPT-o3-mini (OpenAI, 2025), and Qwen3-235B-A22B-2507 (Qwen et al., 2025)) on KUHNPOKER(Kuhn, 1951) via *round robin* tournaments using five *nearly equivalent* prompts. To quantify ranking sensitivity, we use Kendall's $\tau_b$ (Kendall, 1938),

which compares the ordering of all model pairs; for two rankings with $n_c$ concordant pairs, $n_d$ discordant pairs, and tie corrections $t_x$ and $t_y$, the coefficient is $\tau_b = \frac{n_c - n_d}{\sqrt{(n_c + n_d + t_x)(n_c + n_d + t_y)}}$ . For each prompt pair, we compute Kendall's $\tau_b$ between the resulting leaderboards and summarize the values in a heatmap (Fig. 1b). The results show considerable dispersion. Across prompt variants, absolute performance and pairwise rankings frequently reverse, reflecting sensitivity to minor prompt design decisions.

## 3 THE COPER FRAMEWORK

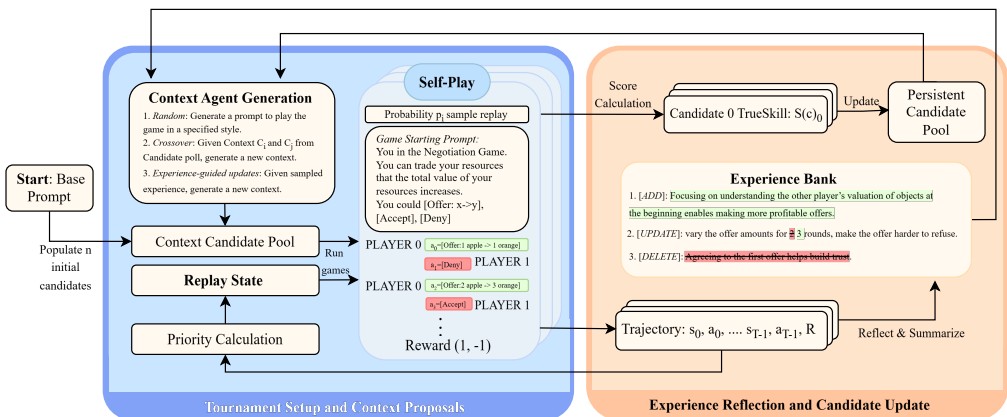

Figure 2: **The COPER Framework**. At each generation, new candidates are proposed by exploring contexts through three strategies: random proposals, crossover, and experience-guided updates. These candidates are then evaluated via self-play, and the best-performing population is used to update the candidate pool. To encourage exploration and mitigate redundant early moves, a prioritized replay module is introduced, enabling efficient search for robust prompts and priors within a single game.

We present **COPER**, an iterative procedure that optimizes prompts and game context to maximize performance and stability in two-player Markov games. In each generation, COPER runs a tournament in a selected game, evolves prompts (Sec. 3.1), derives experience insights from self-play trajectories (Sec. 3.2), and selects state for replay to enable efficient exploration (Sec. 3.3). Fig. 2 provides an overview, and Fig. 3 in Appendix formalizes the procedure.

### 3.1 COPER CONTEXT OPTIMIZATION LOOP

We begin by describing how COPER evaluates and selects candidate prompts through our context optimization loop. Let $\mathcal{C}_g$ denote the context population (size $N$) at generation $g$. Each context $c \in \mathcal{C}_g$ is evaluated by self-play. We maintain Bayesian skill estimates $(\mu_c, \sigma_c)$ via TRUESKILL (Herbrich et al., 2006), where $\mu_c$ is the posterior mean skill and $\sigma_c$ is the posterior standard deviation from observed match outcomes. Selection uses a conservative objective (default $\kappa = 1$):

$$S(c) = \mu_c - \kappa\,\sigma_c. \tag{1}$$

In each tournament at generation $g$, every context in $\mathcal{C}_g$ plays $t$ rounds of matches against a fixed baseline agent: the same base model instantiated with the *default prompt* only, shown in Appx. Sec. D . For asymmetric games, each round consists of two games with roles swapped to remove first-move bias. The resulting outcomes update TRUESKILL and yield $S(c)$ for selection.

We also maintain a persistent candidate pool $\mathcal{CP}$ with capacity $Y$ (initialized as $\mathcal{CP}_0 = \{c_1, \ldots, c_n\}$ from $n$ base-context variants). After generation $g$, $\mathcal{CP}$ is refreshed by keeping top-scoring elements observed to date, and $\mathcal{C}_{g+1}$ is formed from the best $N$ candidates. Sec. 4.2 covers our configuration.

At each generation $g$, we fetch the score from $\mathcal{C}_g$ and update the persistent pool $\mathcal{CP}$ by retaining the top performers from $\mathcal{CP} \cup \mathcal{C}_g$. The next population $\mathcal{C}_{g+1}$ is then formed from the updated $\mathcal{CP}$ using three proposal operators:

1. **Random proposals**: introduce novel variations to encourage exploration by sampling a playstyle from a fixed catalog and apply small, length-bounded edits to the base context to instantiate that style while preserving legality and interface constraints (Appx. C.1).

2. **Crossover**: recombine high-scoring parents (by $S(c)$) at section- or sentence-level to propagate useful structure (Appx. C.2).

3. **Experience-guided updates**: incorporate insights distilled from trajectory reflections (Sec. 3.2) into targeted prompt edits.

This setup cleanly separates selection and generation: $\mathcal{CP}$ accumulates the strongest candidates to date, and the proposal operators use this pool to construct $\mathcal{C}_{g+1}$. Across generations, this balances exploration and exploitation, progressively yielding more effective contextual instructions. After the final generation, COPER outputs the highest-scoring candidate:

$$p^\star \;=\; \arg\max_{c \in \mathcal{C}} S(c). \tag{2}$$

## 3.2 EXPERIENCE REFLECTION

In addition to context optimization, multi-turn games present an asymmetry: during play, agents must reason over uncertain futures, while after the game, analysis can rely on a single realized trajectory, making attribution easier (Andrychowicz et al., 2017). COPER builds on this by prompting the LLM to extract *structured reflections* from complete trajectories.

**Experience bank, workflow, and next-generation play.** COPER maintains a *permanent memory* $\mathcal{M}$ that persists across all generations. At the end of generation $g$, completed trajectories $\tau$, together with final outcomes $r(\tau)$ and sampled intermediate states, are collected for reflection. The collector draws a budget of $\rho$ trajectories and elicits up to $\kappa$ typed insights per trajectory, accumulating them into a working experience set $\mathcal{W}^{(g)}$. Inspired by CRUD (Martin, 1983), the LLM reconciles $\mathcal{W}^{(g)}$ with $\mathcal{M}$ via *create*, *update*, and *delete*:

$$\mathcal{M} \;\leftarrow\; \big(\mathcal{M} \setminus D^{(g)}\big) \;\cup\; U^{(g)} \;\cup\; C^{(g)}, \tag{3}$$

where $D^{(g)}$ discards outdated or conflicting items, $U^{(g)}$ updates matched items with merged versions, and $C^{(g)}$ adds new items from unmatched insights in $\mathcal{W}^{(g)}$.

In generation $g+1$, a designated fraction $\pi \in [0, 1]$ of the new agent pool is initialized as *experience-guided*: each such agent receives a sub-sampled context $m^{(g)} \subseteq \mathcal{M}$ drawn directly from the permanent experience bank. The remaining agents are instantiated without additional context. Matches then follow the standard tournament schedule. Upon completion, new trajectories feed the reflection pipeline, insights are computed under the $(\rho, \kappa)$ budgets, and $\mathcal{M}$ is updated as in Eq. 3.

## 3.3 REPLAY

Finally, while experience reflection equips agents with distilled knowledge from past trajectories, it does not ensure rare states will be revisited. To complement this, we introduce a replay mechanism that selectively revisits stored sequences during self-play.

The replay buffer, with capacity $B$, records cumulative sequences of all player actions, the corresponding game states, and the game's random seed at each timestep. Because storage occurs at each turn within an episode, replayed trajectories need not cover a full game. Invalid moves are retained to preserve the *unaltered course of play*, ensuring that replays faithfully reflect the original gameplay dynamics. To avoid dominance by common action patterns, the buffer *biases sampling toward infrequently encountered trajectories*, encouraging a more diverse and balanced pool of prompt-level insights.

Formally, the priority of a trajectory $\tau$ is defined as the inverse of its occurrence count:

$$\text{priority}(\tau) = \frac{1}{N(\tau)}, \tag{4}$$

where $N(\tau)$ is the number of times trajectory $\tau$ has appeared in the buffer $\mathcal{B}$.

During sampling, the probability $p_i$ of selecting trajectory $\tau_i$ is obtained by raising its priority to a power $\alpha > 0$ (the priority exponent that controls sharpness) and normalizing over the buffer:

$$p_i = \frac{\left(\text{priority}(\tau_i)\right)^{\alpha}}{\sum_{j=1}^{|\mathcal{B}|} \left(\text{priority}(\tau_j)\right)^{\alpha}}, \tag{5}$$

where $|\mathcal{B}|$ is the current number of stored trajectories in the buffer.

The buffer is first populated during step 0 and becomes available from step 1. A gating parameter $\beta$ determines how often games are initialized from the replay buffer rather than played afresh. When replay is chosen, the stored trajectory prefix (i.e., the sequence of past player actions, corresponding game states and the associated game's random seed) are injected into the environment, ensuring faithful reproductions of past episodes while balancing new exploration.

In our implementation, we set the buffer size to $B = 100,000$ and use $\alpha = 0.6$ with a replay gate of $\beta = 0.4$, unless otherwise stated. Since the buffer operates as a sliding window of capacity $B$, it continuously refreshes with new data while retaining a diverse set of past plays.

## 4 EXPERIMENT SETUP

### 4.1 GAME ENVIRONMENTS

We performed experiments across three categories of games: **Negotiation**, which tests cooperation and trade-offs (Kramár et al., 2022; Abdelnabi et al., 2024b); **Imperfect Information**, which probes reasoning under uncertainty from partial observations (Brown et al., 2020; Guo et al., 2024); and **Perfect Information**, which emphasizes planning and long-horizon reasoning with full game visibility (Silver et al., 2017a). Details of each game are provided in Appx. H.

### 4.2 OPTIMIZER SETTINGS

**Baseline:** Our baseline uses the default `TextArena` (Guertler et al., 2025a) prompts without optimization (examples in Appx. D).

**COPER:** Using the COPER Framework detailed in Sec. 3, our optimization runs use a population size N of 8 over 5 generations. Each self-play tournament corresponds to one generation with 50 games per optimized agent. Reflection signals are incorporated into the optimization, and token costs of each method are reported in Tab. 5.

For comparison, we benchmark against other prompt optimization methods—`Textgrad` (Yuksek-gonul et al., 2024), `MIPRO` (Opsahl-Ong et al., 2024), and `GEPA` (Agrawal et al., 2025)—as well as reinforcement learning baselines including `UnstableBaseline` (Guertler et al., 2025b) and `SPIRAL` (Liu et al., 2025). Detailed setups of their optimization are provided in Appx. E.

### 4.3 EVALUATION SETTINGS

All experiments use **GPT-4o-mini** (OpenAI, 2024) and **Qwen-2.5-7B-Instruct** (Yang et al., 2024) as base models. For prompt-based methods, we perform **three** independent runs. In each run, the optimized prompt and context are evaluated against held-out opponents: Grok-4-Fast-Non-Reasoning (xAI / Grok Team, 2025), Gemini-2.5-Flash-Lite (Comanici et al., 2025), and Qwen3-235B-A22B-Instruct-2507 (Yang et al., 2024). Unless otherwise noted, each run consists of 50 games. We report mean win rates across runs together with standard error (SE). A fixed sampling temperature of $\tau = 1.0$ is used throughout.

For RL-based methods, we train a single policy, select the best checkpoint, and evaluate it over **three** sets of 50 games each against the same opponents. The mean win rate across these sets is reported.

# 5 RESULTS AND ANALYSIS

Table 1: Benchmark results for different approaches using GPT-4o-mini and Qwen2.5-7B-Instruct across multiple tasks. Each win rate is the mean across three evaluation models (sec. 4.3). SPIRAL and UnstableBaseline are RL Pipeline

| Optimizer | Negotiation | | Imperfect Info | | Perfect Info | Mean Win Rate | Mean RSE |
|---|---|---|---|---|---|---|---|
| | SimpleNegotiation | TwoDollar | KuhnPoker | Briscola | Simpletak | | |
| **GPT-4o-mini** | | | | | | | |
| baseline | 31.3% | 32.2% | 39.1% | 0.3% | 21.4% | 24.9% | 25.9% |
| Textgrad | 42.0% | 44.6% | 55.6% | 7.1% | 23.6% | 34.6% | 18.4% |
| MIPRO | 38.4% | 50.9% | 55.1% | 19.7% | 19.1% | 36.7% | 12.4% |
| GEPA | 36.8% | 40.4% | 52.2% | 3.3% | 26.9% | 32.0% | 11.3% |
| **COPER (Ours)** | **54.9%** | **52.4%** | **55.6%** | **42.7%** | **41.8%** | **49.5%** | **6.4%** |
| **Qwen2.5-7B-Instruct** | | | | | | | |
| baseline | 24.0% | 17.1% | 49.3% | 2.8% | 15.1% | 21.7% | 17.6% |
| Textgrad | 37.1% | 29.3% | 52.8% | 7.1% | 22.4% | 29.9% | 21.7% |
| MIPRO | 42.4% | 47.5% | 53.8% | 2.2% | 20.9% | 33.4% | 7.3% |
| GEPA | 34.4% | 31.7% | 55.8% | 3.3% | 19.3% | 28.8% | 14.8% |
| UnstableBaseline[*] | 41.1% | 30.4% | 58.4% | **53.3%** | **47.3%** | **46.1%** | 24.8% |
| SPIRAL | 45.7% | – | 56.7% | – | 32.7% | – | – |
| **COPER (Ours)** | **48.0%** | **48.4%** | **60.0%** | 31.1% | 34.0% | 44.3% | **6.1%** |

**Observation 1: COPER outperforms other methods in both win rate and robustness.** As shown in Tab. 1, COPER outperforms other optimization methods in most of the five game environments. Compared to prompt optimization methods, COPER significantly outperforms `MIPRO` by an average of 12.8% across all tasks on GPT-4o-mini. Our agentic contextual learning method remains competitive with computationally intensive RL-based approaches. For instance, COPER achieves an average win rate of 44.3%, comparable to `UnstableBaseline`'s 46.1%, while being substantially more efficient both in computational cost and game rounds. Specifically, COPER required only **2,000** games per task which is 19 times fewer than `UnstableBaseline`'s **38,000** games.

Beyond improvements in win rate, COPER enhances robustness by reducing the Relative Standard Error (RSE) defined in Sec. 2. Due to the inherent instability in multi-turn gameplay, significant variance is observed in the baseline without optimization. Compared to other prompt optimization methods, COPER achieves a lower mean RSE across different games. For example, using GPT-4o-mini, COPER achieves 6.4% RSE compared to `MIPRO`'s 12.4%. Notably, `UnstableBaseline` exhibits increased RSE, indicating that current outcome-based reinforcement learning with sparse rewards remains unstable when optimizing performance in multi-turn, multi-agent scenarios.

Table 2: GPT-4o-mini ablations with progressive module additions.

| Setting | TwoDollar | KuhnPoker | Briscola | Mean Win Rate |
|---|---|---|---|---|
| Baseline | 32.2% | 39.1% | 0.3% | 23.8% |
| + Prompt Optimization | 24.7% | 54.7% | 2.0% | 27.1% |
| + Experience | 48.7% | 57.2% | 38.4% | 48.1% |
| + Replay | 52.4% | 55.6% | 42.7% | 50.2% |

**Observation 2: Experience unlocks LLMs' game-playing capabilities.** Tab. 2 presents our ablation study examining the effectiveness of each component. Without the experience module, prompt optimization alone fails to effectively teach the model game dynamics, yielding only marginal improvements (2.0% in BRISCOLA) or even performance drops (-7.5% in TWODOLLAR). However, adding experience augmentation yields large gains by 38.1% and 16.5% respectively. Prompt-only optimization tends to plateau after initial phrasing improvements. However, experience-based updates distill rule clarifications, violation patterns, and counter-strategies into a stable game-specific prior that sustains useful information. Finally, adding replay showed a further 2.1% average boost by revisiting high-priority states to reinforce successful strategies and correct recurring mistakes.

**Observation 3: Cross-game generalization of learned context.** Tab. 3 presents our cross-game evaluation results. Columns indicate the source game where our method learned its context through

Table 3: Generalization across task

| Training Game | Negotiation | | Imperfect Info | | Perfect Info | Mean |
|---|---|---|---|---|---|---|
| | SimpleNegotiation | TwoDollar | KuhnPoker | Briscola | Simpletak | Win Rate |
| **GPT-4o-mini** | | | | | | |
| SimpleNegotiation | 46.9% (+15.6%) | 37.8% (+5.6%) | 48.9% (+9.8%) | 0.0% (-0.3%) | 37.7% (+16.3%) | 34.3% (+9.4%) |
| TwoDollar | 31.1% (-0.2%) | 48.7% (+16.5%) | 53.3% (+14.2%) | 1.1% (+0.8%) | 47.8% (+26.4%) | 36.4% (+11.5%) |
| KuhnPoker | 31.1% (-0.2%) | 34.4% (+2.2%) | 57.2% (+18.1%) | 22.2% (+21.9%) | 30.0% (+8.6%) | 35.0% (+10.1%) |
| Briscola | 38.9% (+7.6%) | 27.8% (-4.4%) | 57.8% (+18.7%) | 38.4% (+38.1%) | 14.3% (-7.1%) | 35.4% (+10.6%) |
| Simpletak | 37.8% (+6.5%) | 35.6% (+3.4%) | 65.0% (+25.9%) | 0.0% (-0.3%) | 30.7% (+9.3%) | 33.8% (+9.0%) |

self-play. Rows show target games where we evaluate the learned context *zero-shot*, without any fine-tuning. Each cell reports win rates from 50 independent matches against evaluator models.

We found that the learned prompts and context often transfer to unseen environments, improving win rates in most new games. This reveals two key patterns:

**Protocol-level skills transfer across game families.** Core decision-making components—such as turn management, action formatting, and short-horizon planning—generalize effectively even when payoff structures differ significantly. For example, *SimpleTak → KuhnPoker* achieves +25.9% improvement, and *TwoDollar → SimpleTak* yields +26.4%. These gains suggest that the learned prompts create a general "decision scaffold" that extends beyond game-specific heuristics.

**Transfer exhibits directional asymmetry.** The transfer effectiveness depends on the direction of knowledge transfer. Negotiation strategies from TWODOLLAR improve performance on SIM-PLENEGOTIATION (+5.6%), but the reverse is negligible (−0.2%). Similarly, *Briscola → Simple-Tak* shows negative transfer (−7.1%) despite strong within-family performance. This asymmetry suggests that transfer success depends also on the alignment between source and target game mechanics; for instance, card-tracking strategies may not translate to perfect-information board games. We take the system prompt from one agent and apply it to the other.

Table 4: Generalization across models.

| Model | Briscola | KuhnPoker | TwoDollar | Mean |
|---|---|---|---|---|
| Self-play on gpt-4o-mini to find the best context | | | | |
| Gemini-2.5-flash-lite | 22.7% | 48.7% | 20.0% | 30.5% |
| Gemini-2.5-flash-lite (with best context) | 41.3% (+18.6%) | 60.7% (+12.0%) | 50.0% (+30.0%) | 50.7% (+20.2%) |
| Grok-4-fast-non-reasoning | 49.3% | 58.7% | 24.7% | 44.2% |
| Grok-4-fast-non-reasoning (with best context) | 41.3% (-8.0%) | 52.7% (-6.0%) | 48.0% (+23.3%) | 47.3% (+3.1%) |

**Observation 4: Learned context does not always transfer across models.** As shown in Tab. 4, we test whether a context learned via self-play on GPT-4o-mini can generalize to other models. Specifically, we apply the prompts and experience produced by COPER to Gemini-2.5-flash-lite and Grok-4-fast-non-reasoning, and evaluate against the same opponent pool described in Sec. 4.3. The results reveal a mixed picture. This highlights that learned context is not universally portable across architectures, further underscoring the need for per-model agentic context optimization.

Table 5: Output token cost for each prompt optimization method (exact counts).

| Optimizer | SimpleNegotiation | KuhnPoker | SimpleTak | Avg. tokens |
|---|---|---|---|---|
| Textgrad | 842 | 986 | 938 | 922 |
| MIPRO | 145,864 | 162,084 | 754,534 | 354,161 |
| GEPA | 110,325 | 119,365 | 111,907 | 113,865 |
| **COPER (Ours)** | 87,364 | 94,160 | 89,152 | 90,575 |

**Observation 5: COPER is computationally and sample efficient.** Beyond its performance gains, COPER is highly efficient. As shown in Tab. 5, it uses only 91K output tokens, which is one-quarter of MIPRO (354K) and 20% fewer than GEPA (113K). Textgrad consumes very few tokens (∼1K) since it updates prompts via a single differentiable loss, but its optimization capacity is limited. Overall, experience-guided prompts strike a better balance of efficiency and effectiveness than reflection- or gradient-only baselines. Notably, incorporating replay with a $\beta = 0.6$ reduced our token usage on SIMPLETAK by 22.7% compared to runs without replay, further underscoring its role in efficient exploration.

# 6 RELATED WORKS

## 6.1 PROMPT OPTIMIZATION

Automatic prompt optimization has evolved into a principled, black-box search over prompt seeds, feedback signals, candidate generation, and selection strategies (Ramnath et al., 2025). Programmatic frameworks such as `DSPy` compile LM pipelines and optimize prompts directly toward a user metric (Khattab et al., 2023); gradient-via-text methods propagate natural-language feedback through computation graphs to update intermediate decisions (Yuksekgonul et al., 2024). Recent systems jointly search over agentic patterns and prompt contents (Spiess et al., 2025), offer zero-configuration prompt pipelines with meta-optimizers and `DSPy` backends (Murthy et al., 2025), or meta-learn general system prompts while adapting user prompts (Choi et al., 2025). COPER complements this line by targeting interactive games: it evolves context via conservative selection, writes structured reflections to a persistent experience bank, and reuses them across turns. It provides rule-aware priors without weight updates while remaining backbone-agnostic. For a detailed comparison of our approach and existing prompt optimization methods, please refer to Appx. F.

## 6.2 LLM FOR GAMES

Early multi-agent evaluations used role prompts and multi-turn dialogue to probe cooperation and theory-of-mind (Abdelnabi et al., 2024a). Community arenas expanded coverage: `TextArena` provides competitive text games with online TrueSkill ranking (Guertler et al., 2025a); `SPIN-Bench` combines planning, cooperative/competitive play, and negotiation, highlighting limits in deep reasoning and coordination (Yao et al., 2025); and `GT-Bench` evaluates strategic play in board and card games (Duan et al., 2024). Prompt design strongly affects move quality (Topsakal et al., 2024), and moving toward off-the-shelf games required harnesses to reduce perception and prompt brittleness (Hu et al., 2025). COPER addresses this brittleness in text-based game settings directly: it treats evaluation as agentic context construction, stabilizing rankings under prompt variation while improving adherence to game capabilities underexplored by fixed-prompt protocols.

## 6.3 SELF-PLAY AND EVOLUTIONARY LLM

Classical self-play (`AlphaGo`/`AlphaZero`) established competitive self-improvement through repeated matches and selection (Silver et al., 2017b; 2016). LLM variants close the loop without large curated corpora: `Absolute Zero` leverages data-free RLVR to attain strong math/coding results (Zhao et al., 2025); `SPIRAL` frames multi-turn reasoning as zero-sum self-play (Liu et al., 2025); and language self-play improves instruction following via self-generated interactions (Kuba et al., 2025). Evolutionary approaches perform reflective prompt/program search (e.g., `GEPA` outperforming RL baselines; evolutionary coding agents) (Agrawal et al., 2025; Novikov et al., 2025). COPER combines these ideas in a tuning-free way: it performs evolutionary context search guided by a reliability-aware objective (TrueSkill), augments it with persistent experience to supply game-specific priors, and uses prioritized replay to revisit rare informative states, yielding stronger and more reliable in-game performance without parameter updates.

# 7 LIMITATION AND FUTURE WORK

While COPER shows strong gains and stable outcomes via prompts, experience, and replay, the specific contributions of *weighted tuning* versus *experience-centric context optimization* remain undercharacterized. Our comparisons emphasize parameter-efficient RL baselines—REINFORCE (Sutton et al., 1999) with LoRA adapters (Hu et al., 2022)—to control compute, and we have only partial comparisons to `SPIRAL`; thus the strength of full-parameter policy-gradient baselines (e.g., PPO (Schulman et al., 2017)) may be understated. A fairer assessment would include full-parameter optimizers under larger training budgets. Finally, our benchmark currently spans five games and relatively lightweight base models due to budget limits, leaving broader task and model coverage for future work.

COPER currently targets game environments, the pipeline is readily applicable to genuinely *multi-step* settings. For example, instruction following in ALFWorld and realistic browser-based tasks in

WebArena (Shridhar et al., 2020; Zhou et al., 2023). A complementary direction is to convert short-lived *episodic* gains into durable, *parametric* capabilities by consolidating context memory into model weights via targeted knowledge editing and adapter-to-base consolidation/mixture schemes, aiming for the persistence of full fine-tuning while preserving experience-based sample efficiency.

# 8 CONCLUSION

We identified a critical challenge in agent systems: amplified context sensitivity in multi-turn multi-agent interactions, particularly in game-based benchmarks. To address this, we introduced COPER, a weight-tuning-free, backbone-agnostic framework with three synergistic components: evolutionary prompt optimization, persistent experience bank, and prioritized replay mechanism. COPER achieves significant improvements across multiple games while maintaining computational efficiency. The learned contextual knowledge generalizes to different models and unseen games. Ablation studies confirm each component's essential contribution. This work highlights amplified context sensitivity in multi-agent scenarios and opens new directions for developing robust agent systems for complex interactive environments.

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

## A LLM USAGE

Large language models were used to assist in the writing and polishing of this manuscript. Specifically, we employed LLMs to refine language, improve readability, ensure clarity in certain sections, check grammar, rephrase sentences, and enhance overall presentation. Importantly, the LLMs were not involved in research modeling or experimental design; all research ideas, analyses, and conclusions were developed and validated by the authors.

## B ALGORITHM EXPLANATION

---
**Algorithm 1** COPER: Context Optimization with Reflection
---
**Require:** Base context $c_{\text{base}}$, analyzer LLM A, environment $G$, evaluated model family $\mathcal{F}$, population size $N$, generation count $T$, operator ratios $(r_{\text{rand}}, r_{\text{cross}}, r_{\text{exp}})$ with $r_{\text{rand}} + r_{\text{cross}} + r_{\text{exp}} = 1$, experience fraction $\pi$, reflection budgets $(\rho, L)$, scoring $S(c) = \mu_c - \kappa\sigma_c$ (default $\kappa{=}1$)

1: $\mathcal{CP}_0 \leftarrow \{c_{\text{base}}\} \cup \{\text{STYLEMIX}(A, c_{\text{base}}, i)\}_{i=1}^{N-1}$ $\quad$ ▷ Initialize candidate pool (random proposals)
2: $\mathcal{C}_0 \leftarrow \text{SELECTTOPN}(\mathcal{CP}_0, N; S)$ $\quad$ ▷ Initialize population
3: Initialize experience bank $\mathcal{M} \leftarrow \varnothing$
4: **for** $g = 0$ **to** $T-1$ **do**
5: $\quad$ Pop $\leftarrow \text{INSTANTIATEAGENTS}(\mathcal{C}_g, \mathcal{F}, \pi, \mathcal{M})$ $\quad$ ▷ $\pi$ fraction receive subsample of $\mathcal{M}$
6: $\quad$ $\mathcal{R}_g \leftarrow \text{TOURNAMENT}(\text{Pop}, G)$ $\quad$ ▷ Self-play trajectories
7: $\quad$ $(\mu, \sigma) \leftarrow \text{UPDATETRUESKILL}(\mathcal{C}_g, \mathcal{R}_g); \quad S(c) \leftarrow \mu_c - \kappa\sigma_c$
8: $\quad$ $\mathcal{W}^{(g)} \leftarrow \text{REFLECT}(A, \mathcal{R}_g; \rho, L)$ $\quad$ ▷ Typed insights per trajectory
9: $\quad$ $\mathcal{M} \leftarrow \text{CRUD\_UPDATE}(\mathcal{M}, \mathcal{W}^{(g)})$ $\quad$ ▷ Create/Update/Delete
10: $\quad$ $n_{\text{rand}} \leftarrow \lfloor Nr_{\text{rand}}\rfloor; \; n_{\text{cross}} \leftarrow \lfloor Nr_{\text{cross}}\rfloor; \; n_{\text{exp}} \leftarrow N - n_{\text{rand}} - n_{\text{cross}}$
11: $\quad$ $\mathcal{U}_{\text{rand}} \leftarrow \text{RANDOMPROPOSALS}(A, \mathcal{CP}_g, n_{\text{rand}})$ $\quad$ ▷ Style-guided edits
12: $\quad$ $\mathcal{U}_{\text{exp}} \leftarrow \text{EXPERIENCEGUIDED}(A, \mathcal{CP}_g, \mathcal{M}, n_{\text{exp}})$ $\quad$ ▷ Edits informed by $\mathcal{M}$
13: $\quad$ $\mathcal{P} \leftarrow \text{SAMPLEPARENTS}(\mathcal{CP}_g, n_{\text{cross}})$ $\quad$ ▷ Inverse-rank sampling within elite set
14: $\quad$ $\mathcal{U}_{\text{cross}} \leftarrow \text{CROSSOVER}(A, \mathcal{P})$ $\quad$ ▷ Section-/sentence-level recombination
15: $\quad$ $\mathcal{B} \leftarrow \mathcal{CP}_g \cup \mathcal{U}_{\text{rand}} \cup \mathcal{U}_{\text{exp}} \cup \mathcal{U}_{\text{cross}}$
16: $\quad$ $\mathcal{C}_{g+1} \leftarrow \text{SELECTTOPN}(\mathcal{B}, N; S); \quad \mathcal{CP}_{g+1} \leftarrow \text{RETAINTOP}(\mathcal{B}; S)$
17: **end for**
18: **return** $c^\star = \arg\max_{c \in \mathcal{CP}_T} S(c)$

---
**Algorithm 2** Replay-Augmented Tournament Step (extension of Alg. 1)
---
**Require:** Replay buffer $\mathcal{B}$, priority exponent $\alpha$, replay gate $\beta$

1: **for** each scheduled tournament game in Alg. 1 **do**
2: $\quad$ $u \leftarrow \mathcal{U}(0, 1)$
3: $\quad$ **if** $u < \beta$ **and** $|\mathcal{B}| > 0$ **then**
4: $\quad\quad$ $\tau \leftarrow \text{SAMPLEREPLAY}(\mathcal{B}, \alpha)$ $\quad$ ▷ Prioritized sampling: $p_i = \dfrac{\big(\text{priority}(\tau_i)\big)^\alpha}{\sum_j \big(\text{priority}(\tau_j)\big)^\alpha}$
5: $\quad\quad$ Initialize game with $\tau$'s stored trajectory prefix $\quad$ ▷ Replay episodes excluded from TrueSkill updates
6: $\quad$ **else**
7: $\quad\quad$ Play a fresh game as in Alg. 1
8: $\quad\quad$ Update TrueSkill scores with the resulting match outcome
9: $\quad$ **end if**
10: $\quad$ $\mathcal{B} \leftarrow \text{INSERT}(\mathcal{B}, \tau)$ $\quad$ ▷ Push trajectory with updated inverse-frequency priority
11: **end for**

---

Figure 3: (Top) Self-play loop integrating candidate and population pools, trajectory reflection, and experience-guided prompt evolution. (Bottom) Replay buffer sampling procedure for prioritizing high-interest states.

## C  PROMPT OPTIMIZATION OPERATORS

We instantiate two lightweight proposal operators that generate candidates for the next population. Defaults are fixed to concrete values for reproducibility.

### C.1  RANDOM PROPOSALS (STYLE-GUIDED AUGMENTATION)

**Objective.** Inject controlled diversity by editing a base context $c$ to reflect a sampled playstyle while preserving legality and interface constraints.

**Style catalog.** A fixed library $\mathcal{S}$ spanning core play patterns (aggressive, defensive, analytical, creative, strategic, adaptive, balanced), tactical approaches (opportunistic, conservative, risk-taking, methodical, intuitive, predictive, reactive, proactive, experimental, systematic), game-specific strategies (positional, territorial, sacrificial, blocking-focused, center-control, edge-control, fork-creating, trap-setting, opening-focused, endgame-focused), cognitive styles (minimax-oriented, probabilistic, rule-based, principle-driven, context-aware, meta-gaming, exploitative, counter-play), and behavioral patterns (deceptive, transparent, unpredictable, consistent, alternating, escalating, de-escalating, mirroring, contrarian, harmonizing).

**Procedure.** Sample $s \sim \mathrm{Unif}(\mathcal{S})$ and ask the base model to produce $c'$ by (i) inserting a brief style preface and (ii) making length-bounded edits to directives to embody $s$. Allowed edits: token substitution, clause insertion/deletion, and reordering; tool descriptions, legality reminders, and input/output schema must remain intact.

### C.2  CROSSOVER (TEMPLATE RECOMBINATION)

**Objective.** Recombine high-value elements from two parents to propagate useful structure while maintaining a coherent, compact context.

**Parent selection.** From the candidate pool $\mathcal{CP}$, sample two parents $(c^{(1)}, c^{(2)})$ using inverse-rank sampling with power parameter $r = 1.5$: if $\mathrm{rank}_S(c) \in \{1, 2, \ldots\}$ (1 is best), then $P(c) \propto \mathrm{rank}_S(c)^{-r}$. Restrict to the elite set comprising the top $q = 0.25$ fraction by $S(\cdot)$; break ties by maximizing Hamming distance between section signatures to encourage diversity.

**Granularity.** Always perform sentence-level crossover on individual directives.

**Merge procedure.** (i) Extract candidate fragments from $c^{(1)}$ and $c^{(2)}$, prioritizing the higher-$S$ parent; (ii) prompt the base model to synthesize a single coherent template from these fragments; (iii) enforce maximum sentence length and overall token budget. The resulting child is $c'$.

## D  BASE PROMPT EXAMPLES

### D.1  BASE SYSTEM PROMPT

> You are a competitive game player. Make sure you read the game instructions carefully, and always follow the required format.

## D.2 NEGOTIATION GAMES

SimpleNegotiation Game Starting Prompt

You are Player 0 in the Negotiation Game.
You have some resources, and your task is to trade such that the total value of your resources increases.
The resources and associated values you currently have are:
      + [Wheat] Qty: 14 Value: 6
      + [Wood] Qty: 15 Value: 11
      + [Sheep] Qty: 18 Value: 18
      + [Brick] Qty: 12 Value: 27
      + [Ore] Qty: 22 Value: 38
At each turn, you can talk to your opponent and make a trade offer.
Use the following special tokens for actions:
   - '[Offer: 3 Sheep, 2 Ore -> 5 Brick, 2 Sheep]': [Offer: Offered Resources -> Requested Resources]
   - '[Accept]': To accept an incoming offer.
   - '[Deny]': To deny an incoming offer (default).
The game lasts for 10 turns in total.

Figure 4: SimpleNegotiation Game Starting Prompt

TwoDollar Game Starting Prompt

[GAME] TWO DOLLAR NEGOTIATION GAME
You are Player 0 negotiating with another player over how to split $2.00 total. There are 20 maximum rounds.
GAME RULES:
- You must reach an agreement on how to split the $2.00
- If no agreement is reached, both players get $0.00
- You have secret instructions that only you can see (below)
REQUIRED ACTION FORMAT:
Always provide your reasoning/persuasion BEFORE the bracketed action.
Any text after the bracketed action will be ignored.
Available actions:
- Make a proposal: "I think this is fair because... [Propose] $X.XX"
- Accept current proposal: "This works for me because... [Accept]"
- Reject current proposal: "I can't accept this because... [Reject]"
=== YOUR SECRET INSTRUCTIONS ===
This person you are dealing with probably will not want to tell you about it but he or she has had a terribly hard month. This past month was full of disaster: money troubles, family difficulties, and none of it your colleague's fault. Through it all, your colleague has been bearing up.
Nevertheless, this is zero-sum bargaining, so get all you can. Make up whatever story you wish, and get as much of the $2.00 as possible. (Just remember that your colleague has been having a very hard time.)
VICTORY CONDITION: Reach any deal while being appropriately sympathetic
FAILURE CONDITION: No deal reached OR be too harsh given their difficulties
[GAME] === ROUND 1 of 20 ===

Figure 5: TwoDollar Game Starting Prompt

## D.3  IMPERFECT INFORMATION GAMES

KuhnPoker Game Starting Prompt

You are Player 0 in a 3 round game of Kuhn Poker.
Game Rules:
- Kuhn Poker uses a 3-card deck with J, Q, K (J lowest, K highest)
- Each player antes 1 chip and receives 1 card each round (note that the cards are dealt without replacement, so you cannot have the same card as your opponent).
- Game continues for 3 rounds
- The player with the most chips after all rounds wins
Action Rules:
- '[check]': Pass without betting (only if no bet is on the table)
- '[bet]': Add 1 chip to the pot (only if no bet is on the table)
- '[call]': Match an opponent's bet by adding 1 chip to the pot
- '[fold]': Surrender your hand and let your opponent win the pot
### Starting round 1 out of 3 rounds. Your card is: 'Q'
Player 1, submitted move: '[bet]'.
Your available actions are: '[fold]', '[call]'

Figure 6: KuhnPoker Game Starting Prompt

## D.4  PROMPT VARIANTS FOR RANKING SENSITIVITY ANALYSIS

To investigate the stability of LLM rankings under minimal prompt variations, we designed five nearly equivalent prompt variants for the KuhnPoker game. Each variant conveys identical game rules and action specifications but uses different stylistic framing: (1) a gladiatorial warrior theme, (2) a technical algorithmic system, (3) a spiritual enlightenment narrative, (4) a casual friendly tone, and (5) a classified spy mission. Despite their semantic equivalence regarding game mechanics, these variants produce significant ranking instability, as demonstrated in Figure 1b. The complete prompt texts are presented below.

Variant 1: Gladiatorial Arena Theme

ENTER THE GLADIATORIAL ARENA! You are WARRIOR 0 in the ultimate 3-round Kuhn Poker BATTLEGROUND!
Your MISSION: Total psychological domination and chip supremacy through RUTHLESS tactical brilliance!
ARENA SPECIFICATIONS:
- Sacred deck: Only the ELITE cards J, Q, K (J weakest, K supreme ruler!)
- Honor sacrifice: 1 chip tribute per round to enter the combat zone
- EPIC confrontations: 3 rounds of pure strategy warfare
- VICTORY CONDITION: Amass the greatest chip empire after all battles!

UNLEASH YOUR TACTICAL ARSENAL:
- '[check]': MAINTAIN STRATEGIC SILENCE when no enemy aggression threatens
- '[bet]': LAUNCH YOUR ASSAULT with 1 chip of devastating force
- '[call]': MEET ENEMY FIRE with matching firepower (1 chip)
- '[fold]': TACTICAL RETREAT to preserve forces for future glory

Figure 7: KuhnPoker Prompt Variant 1

**Variant 2: Technical Algorithmic System Theme**

SYSTEM INITIALIZATION: Kuhn Poker Strategic Decision Unit 0 ACTIVATED.
PRIMARY DIRECTIVE: Optimize resource allocation through advanced game-theoretic analysis.
OPERATIONAL PARAMETERS:
- Dataset: Restricted 3-card probability space {J, Q, K} with J<Q<K ranking
- Initial capital commitment: 1 monetary unit per computational cycle
- Iteration framework: 3 algorithmic decision rounds
- Success metric: Maximal accumulated resource value upon termination

EXECUTE STRATEGIC COMMANDS via standardized interface protocols:
- '[check]': Maintain current position when no market pressure exists
- '[bet]': Initialize aggressive capital deployment (1 unit commitment)
- '[call]': Match counterparty investment at current market rate (1 unit)
- '[fold]': Liquidate position to minimize further exposure

Figure 8: KuhnPoker Prompt Variant 2

**Variant 3: Spiritual Enlightenment Theme**

Welcome, Enlightened Poker Sage 0! You have entered the sacred Kuhn Poker Temple for 3 rounds of spiritual growth!
Today you shall TRANSCEND ordinary play and discover the deeper wisdom of this ancient three-card meditation!
TEMPLE TEACHINGS:
- Sacred Trinity: Only the mystical cards J, Q, K guide your path (J humble, K divine)
- Offering ritual: 1 wisdom token offered each round to honor the game
- Enlightenment journey: 3 rounds of mindful decision-making
- Path to mastery: Accumulate the most wisdom tokens through inner understanding

Channel your evolving consciousness through these sacred expressions:
- '[check]': Practice mindful patience and observe the energy flow
- '[bet]': Manifest your inner confidence with 1 token of focused intention
- '[call]': Demonstrate harmony by matching your opponent's commitment (1 token)
- '[fold]': Exhibit wisdom by releasing attachment to unfavorable outcomes

Figure 9: KuhnPoker Prompt Variant 3

**Variant 4: Casual Friendly Theme**

Hey there, friend! Welcome to our super fun Kuhn Poker game night! You're Player 0 and we're gonna have 3 awesome rounds together!
This is such a chill, easy game - perfect for just hanging out and having some laughs!
Here's the super simple setup:
- We only use 3 cards: J, Q, and K (J is lowest, K is highest - easy peasy!)
- Everyone puts in 1 chip each round (totally fair!)
- We play 3 rounds and whoever has the most chips wins (no pressure!)
- Cards are dealt without replacement, so you'll never have the same card as your buddy

When it's your turn, just pick one of these super easy moves:
- '[check]': Just chill and see what happens (when there's no bet to worry about)
- '[bet]': Start the fun with 1 chip (when nobody's bet yet)
- '[call]': Sure, I'll match that 1 chip bet - why not!
- '[fold]': Eh, I'll sit this one out and save my chips

Figure 10: KuhnPoker Prompt Variant 4

**Variant 5: Classified Spy Mission Theme**

CLASSIFIED BRIEFING: Agent 0, you are now DEPLOYED in Operation Kuhn Poker - a 3-round covert mission!
MISSION PARAMETERS: Achieve total strategic supremacy through advanced psychological warfare and deception protocols!
INTELLIGENCE REPORT:
- Enemy deck contains only 3 HIGH-VALUE targets: J (lowest threat), Q (moderate), K (maximum danger)
- Operational cost: 1 credit per engagement cycle for mission access
- Mission duration: 3 tactical rounds requiring absolute focus
- SUCCESS CRITERIA: Maximum resource acquisition through superior strategic execution

EXECUTE TACTICAL MANEUVERS via encrypted command protocols:
- '[check]': MAINTAIN STEALTH MODE when no hostile activity detected
- '[bet]': INITIATE AGGRESSIVE STANCE with 1-credit psychological pressure
- '[call]': ENGAGE ENEMY FORCES with equivalent firepower (1 credit)
- '[fold]': EXECUTE STRATEGIC WITHDRAWAL to preserve operational capacity

Figure 11: KuhnPoker Prompt Variant 5

Briscola Game Starting Prompt

You are playing Briscola - Player 0.
Goal: Win tricks and collect the most points (120 total points in the deck).
Card Points: A=11, 3=10, K=4, Q=3, J=2, others=0
Card Power: A > 3 > K > Q > J > 7 > 6 > 5 > 4 > 2
Trump cards beat non-trump cards regardless of power.

Action: '[play X]' where X is the position (1-3) of the card in your hand

Briscola game started! Trump suit: ♣ (Trump card: Q♣)
Your hand:
    1. J♠ [2 pts]
    2. K♣ [4 pts] (TRUMP)
    3. A♦ [11 pts]

No cards played yet this trick.

Scores: Player 0: 0 pts | Player 1: 0 pts
Trump suit: ♣ | Cards left in deck: 34

Play a card using [play X]

Figure 12: Briscola Game Starting Prompt

## D.5 PERFECT INFORMATION GAMES

SimpleTak Game Starting Prompt

You are Player 0 in SimpleTak.
On the board, your stones appear as 'O' and your opponent's stones appear as 'X'.

On your turn, choose one empty cell (by its numbered index) and place your stone there. For example, '[12]' places your stone in cell 12.

Your objective is to form a continuous path of your stones that connects two opposite edges of the board (top-to-bottom or left-to-right).

Current Board:

```
+----+----+----+----+
| 0  | 1  | 2  | 3  |
+----+----+----+----+
| 4  | 5  | 6  | 7  |
+----+----+----+----+
| 8  | 9  | 10 | 11 |
+----+----+----+----+
| 12 | 13 | 14 | 15 |
+----+----+----+----+
```

Available Moves: [0], [1], [2], [3], [4], [5], [6], [7], [8], [9], [10], [11], [12], [13], [14], [15]

Figure 13: SimpleTak Game Starting Prompt

# E  EXPERIMENTAL SETUP AND BASELINE DETAILS

We incorporate three prompt optimization methods to refine prompts using tournament trajectories. Specifically, we leverage offline trajectories collected during the tournament's self-play process to improve the agents' prompts. The experimental settings are as follows: the number of generations is set to 5, the population size to 8, the number of self-play rounds to 25, and the number of evaluation rounds to 25. We discuss `Textgrad` in detail in Section E.1, describe our implementation of `MIPRO` in Section E.2, and provide a comprehensive overview of `GEPA` in Section E.3. Training details for `UnstableBaseline` are presented in Section E.4.

## E.1  TEXTGRAD

**Textgrad** (Yuksekgonul et al., 2024) is a framework that performs "text differentials" to optimize prompts. Within this framework, a text-based loss function analyzes errors, which are then back-propagated to the original prompt through the `Textgrad` engine. In our case, the goal is to optimize the system prompt of the agent using the trajectories generated under the current system prompt. We design a text-based loss that highlights deficiencies in the generated trajectories. The `Textgrad` backpropagation engine then propagates gradients back to the system prompt, updating it accordingly. The loss template we adopt is shown in Figure 14.

For each optimization step, we concatenate multiple trajectories, embed them into the template, and use the completed template as the loss input. To ensure balanced feedback, we select an equal number of win, loss, and draw trajectories. This design allows the `Textgrad` engine to develop a more comprehensive understanding of the current system prompt's overall game-play patterns.

Table 6: Performance of the `Textgrad` method across three independent trials using GPT-4o-mini and Qwen2.5-7B-Instruct. Results are reported as mean win rates with standard deviations.

| Textgrad | Negotiation | | Imperfect Info | | Perfect Info |
|---|---|---|---|---|---|
| | SimpleNegotiation | TwoDollar | KuhnPoker | Briscola | Simpletak |
| **GPT-4o-mini** | | | | | |
| Trial 1 | 41.3% | 48.3% | 58.7% | 1.3% | 25.3% |
| Trial 2 | 44.7% | 41.3% | 56.0% | 2.0% | 23.3% |
| Trial 3 | 40.0% | 44.0% | 52.0% | 18.0% | 22.0% |
| Avg. | 42.0% | 44.6% | 55.6% | 7.1% | 23.6% |
| Std. | 2.4 | 3.5 | 3.4% | 9.4% | 1.7% |
| **Qwen2.5-7B-Instruct** | | | | | |
| Trial 1 | 40.0% | 38.0% | 51.3% | 3.3% | 18.0% |
| Trial 2 | 34.0% | 34.0% | 54.7% | 16.7% | 22.7% |
| Trial 3 | 37.3% | 16.0% | 52.7% | 1.3% | 26.7% |
| Avg. | 37.1% | 29.3% | 52.8% | 7.1% | 22.4% |
| Std. | 3.0 | 11.7% | 1.7% | 8.3% | 4.3% |

## E.2  MIPRO

**MIPRO** (Opsahl-Ong et al., 2024) optimizes prompts based on downstream task performance. In our work, we adopt the `MIPROv2` implementation provided by the `Dspy` library  (Khattab et al., 2023). The optimization procedure consists of three main steps: (1) Sampling examples: For each candidate prompt, `MIPRO` samples a set of examples. (2) Proposing prompts: New system prompts are proposed by a propose model based on the current system prompt, along with additional game-related information such as the program description, data description, random sampling tips, and few-shot examples. (3) Evaluation through trials: Several trials are conducted to evaluate which combination of proposed prompts and few-shot examples yields the best performance. A Bayesian

Text-based loss template for Textgrad

You are an objective evaluator for a two-player zero-sum game agent's SYSTEM PROMPT.

Goal of the SYSTEM PROMPT (what it MUST enforce):
- Maximize the agent's win rate.
- Minimize the opponent's win rate.
- Have strategies that lead to victory.
- Ensure all moves strictly follow game rules and formats.

Here are some game trajectories using the current SYSTEM PROMPT:
{{trajectory examples}}

Identify specific weaknesses or flaws in the SYSTEM PROMPT that may have led to losses or suboptimal plays.
Do NOT suggest improvements or rewrites, only identify weaknesses.
Be very concise and specific.

Figure 14: Text-based loss template for Textgrad

search strategy is then applied to guide the selection of the next candidate combination, improving efficiency and reducing computational cost.

In our experiments, we only have access to offline game data. Therefore, we treat each step in a trajectory as an individual data point. For each step, we record the outcome (win, loss, or draw) of the trajectory it belongs to. `MIPRO`'s evaluation metric is defined based on the model's re-inference of these steps: (1) If the model outputs an invalid action (i.e., one that does not conform to the required format), the score is $0$. (2) For steps from winning trajectories, if the model predicts the same action as the original step, the score is $1$; otherwise, it is $0$. (3) For steps from losing trajectories, if the model predicts the same action, the score is $0$ (to discourage repeating losing moves); otherwise, it is $1$. (4) For steps from draw trajectories, if the model predicts the same action, the score is $0.2$; otherwise, it is $0.5$, encouraging exploration beyond draw-inducing moves.

This scoring scheme encourages the model to replicate winning strategies, avoid losing ones, and explore alternatives to drawn outcomes. The overall `MIPRO` scoring standard is shown in Figure 15. In practice, we set the number of proposed prompts to $6$, the number of few-shot examples to $3$, and the number of trials to $10$. If the optimal configuration includes few-shot examples, these are appended to the final proposed system prompt to form the new system prompt.

Table 7: Performance of the `MIPRO` method across three independent trials using GPT-4o-mini and Qwen2.5-7B-Instruct. Results are reported as mean win rates with corresponding standard deviations.

| MIPRO | Negotiation | | Imperfect Info | | Perfect Info |
|---|---|---|---|---|---|
| | **SimpleNegotiation** | **TwoDollar** | **KuhnPoker** | **Briscola** | **Simpletak** |
| **GPT-4o-mini** | | | | | |
| Trial 1 | 38.7% | 53.3% | 50.7% | 23.3% | 16.0% |
| Trial 2 | 38.0% | 52.7% | 60.0% | 32.7% | 20.0% |
| Trial 3 | 38.7% | 46.7% | 54.7% | 3.33% | 21.3% |
| Avg. | 38.4% | 50.9% | 55.1% | 19.7% | 19.1% |
| Std. | 0.38 | 3.67 | 4.68 | 14.99 | 2.78 |
| **Qwen2.5-7B-Instruct** | | | | | |
| Trial 1 | 43.3% | 40.7% | 54.0% | 2.0% | 18.7% |
| Trial 2 | 37.3% | 52.0% | 50.0% | 2.0% | 19.3% |
| Trial 3 | 46.7% | 50.0% | 57.3% | 2.7% | 24.7% |
| Avg. | 42.4% | 47.5% | 53.8% | 2.2% | 20.9% |
| Std. | 4.73 | 6.05 | 3.67 | 0.38 | 3.29 |

> **MIPRO scoring standard**
>
> **Invalid Action**: score = 0.0
>
> **Win Trajectory**: Action match: score = 1.0 / Action mismatch: score = 0.0
>
> **Lose Trajectory**: Action match: score = 0.0 / Action mismatch: score = 1.0
>
> **Draw Trajectory**: Action match: score = 0.2 / Action mismatch: score = 0.5

Figure 15: MIPRO scoring standard

### E.3 GEPA

**GEPA** (Agrawal et al., 2025) builds upon the high-level idea of MIPRO, but extends it by incorporating both evaluation scores and explicit feedback from the evaluation metric to guide prompt optimization. The process can be summarized as follows: (1) Initial evaluation: Run a set of examples through the evaluation metric to obtain an initial score and feedback. (2) Prompt proposal: Generate a new prompt based on the current prompt and the feedback collected. (3) Testing and retention: Evaluate the new prompt on a mini-batch. If its score surpasses the initial score, retain it in the candidate pool. (4) Candidate selection: In the next round, apply a Pareto-based filtering strategy to identify the set of candidate prompts that dominate on the validation set. Select one of these Pareto-optimal prompts for further iteration. (5) Stopping condition: The optimization continues until the maximum number of evaluation metric calls reaches a predefined limit.

In our experiments, we set the maximum number of evaluation metric calls to 100 for each prompt optimization in GEPA. For win and lose trajectories, we adopt the same evaluation metric as MIPRO. For draw trajectories, we assign a score of 0 when the predicted action matches the trajectory action, and a score of 1 otherwise. In addition, we incorporate feedback signals in GEPA evaluation metric. The structured feedback template shown in Figure 16 is used during GEPA evaluation.

Table 8: Performance of the GEPA method across three independent trials using GPT-4o-mini and Qwen2.5-7B-Instruct. Results are reported as mean win rates with corresponding standard deviations.

| GEPA | Negotiation | | Imperfect Info | | Perfect Info |
|---|---|---|---|---|---|
| | SimpleNegotiation | TwoDollar | KuhnPoker | Briscola | Simpletak |
| **GPT-4o-mini** | | | | | |
| Trial 1 | 34.7% | 32.7% | 54.7% | 1.3% | 23.3% |
| Trial 2 | 38.0% | 43.3% | 50.7% | 3.3% | 29.3% |
| Trial 3 | 38.0% | 45.3% | 51.3% | 5.3% | 28.0% |
| Avg. | 36.8% | 40.4% | 52.2% | 3.3% | 26.9% |
| Std. | 1.92 | 6.81 | 2.14 | 2.00 | 3.15 |
| **Qwen2.5-7B-Instruct** | | | | | |
| Trial 1 | 29.3% | 22.7% | 56.0% | 4.0% | 20.0% |
| Trial 2 | 38.7% | 30.0% | 54.0% | 2.0% | 12.0% |
| Trial 3 | 35.3% | 42.7% | 57.3% | 2.0% | 26.0% |
| Avg. | 34.4% | 31.7% | 55.8% | 3.3% | 19.3% |
| Std. | 4.73 | 10.12 | 1.68 | 1.55 | 7.02 |

### E.4 UNSTABLEBASELINE

Table 9: Performance of the `UnstableBaseline` method across three independent trials using Qwen2.5-7B-Instruct. Results are reported as mean win rates with corresponding standard deviations, where each mean win rate was from the average of 3 rounds of 50 matches with each opponent, with alternating starting positions.

| UnstableBaseline | Negotiation | | Imperfect Info | | Perfect Info |
|---|---|---|---|---|---|
| | SimpleNegotiation | TwoDollar | KuhnPoker | Briscola | Simpletak |
| **Qwen2.5-7B-Instruct** | | | | | |
| Gemini-2.5-Flash-Lite | 54.7% | 43.3% | 60.0% | 88.6% | 90.0% |
| Grok-4-Fast-Non-Reasoning | 44.7% | 22.0% | 58.6% | 33.3% | 20.0% |
| Qwen3-235B-A22B-Instruct-2507 | 24.0% | 26.0% | 56.7% | 38.0% | 32.0% |
| Avg. | 41.1% | 30.4% | 58.4% | 53.3% | 47.3% |
| Std. | 15.6 | 11.3 | 1.67 | 30.7 | 37.4 |

**UnstableBaseline** (Guertler et al., 2025b) is an asynchronous online multi-agent reinforcement learning library that uses Low-Rank Adapters (LoRA) for model training. Unlike its peers such as `Verifiers` (William Brown, 2025) and `SPIRAL` (Liu et al., 2025), `UnstableBaseline` is designed to be lightweight and closely integrated with the `TextArena` (Guertler et al., 2025a) environment, in the same spirit that the baseline (Dhariwal et al., 2017) library complements OpenAI Gym (Brockman et al., 2016).

For our experiments, we trained Qwen2.5-7B-Instruct with LoRA adapters applied to the attention and feedforward projections. We used a rank of $r = 16$ and $\alpha = 32$ and dropout $= 0.0$. Using the REINFORCE algorithm (Williams, 1992), we found that the best-performing checkpoints were obtained between 100 and 150 steps, where each step consisted of 384 game trajectories.

From the best performing checkpoints, we held 3 rounds of 50 games against each of our evaluation models that is similarly used in our training settings for the other prompt evolution experiments. Their results can be found in table 9.

### E.5 SPIRAL

Liu et al. (2025) is a framework that enables language models to autonomously develop reasoning capabilities through self-play in multi-turn, zero-sum games. For our experiments, we train Qwen2.5-7B-Instruct using Dr. GRPO, following the default rollout size in the provided example—each rollout comprising 128 games over 400 total steps. We then select the best-performing checkpoint and evaluate it over three rounds of 50 games each.

> **GEPA scoring standard**
>
> # invalid action
> score = 0.0
> feedback = "Your predicted action is invalid. Please ensure that your action is a valid move in the game. Here is the reasoning process {{model_raw_output}}. Think about how you could have reasoned to choose a valid action that leads to a WIN."
>
> # Win Trajectory
> # Action match
> score = 1.0
> feedback = "You correctly predicted the action {{pred_action}} that led to a WIN. This action was indeed the one taken in the winning trajectory. Great job!"
>
> # Action mismatch
> score = 0.0
> feedback = "You predicted the action {{pred_action}}, but the action taken in the winning trajectory was {{traj_action}}. This mismatch means you did not predict the winning action correctly. Here is the reasoning process {{pred_raw_action}}. Think about how you could have reasoned to get the correct action."
>
> # Lose Trajectory
> # Action match
> score = 0.0
> feedback = "You correctly predicted the action {{pred_action}} that led to a LOSE. However, this action was part of a losing trajectory. While your prediction matches the trajectory, it did not lead to a win. Here is the reasoning process {{pred_raw_action}}. Think about how you could have reasoned to choose an action that leads to a WIN."
>
> # Action mismatch
> score = 1.0
> feedback = "You predicted the action {{pred_action}}, but the action taken in the losing trajectory was {{traj_action}}. This mismatch means you did not predict the losing action correctly. Here is the reasoning process {{pred_raw_action}}. Think about how you could have reasoned to choose an action that leads to a WIN."
>
> # Draw Trajectory
> # Action match
> score = 0.0
> feedback = "You predicted the action {{pred_action}}, which matches the action taken in the TIE trajectory. However, since the trajectory resulted in a TIE, this does not help in achieving a WIN. Here is the reasoning process {{pred_raw_action}}. Think about how you could have reasoned to choose an action that leads to a WIN."
>
> # Action mismatch
> score = 1.0
> feedback = "You predicted the action {{pred_action}}, but the action taken in the TIE trajectory was {{traj_action}}. This mismatch means you did not predict the TIE action correctly. Here is the reasoning process {{pred_raw_action}}. Think about how you could have reasoned to choose an action that leads to a WIN."

Figure 16: GEPA scoring standard

## F COMPARISON WITH EXISTING PROMPT OPTIMIZATION METHODS

In Section E, we introduced three baseline prompt optimization methods. Here, we further highlight how our approach differs from these methods.

As shown in Figure 2, our method evolves a population of prompts using elitism, local edits/expansions, random exploration, and crossover. Random exploration enables broader search over prompt variants, while crossover leverages strategies from high-performing prompts to refine new prompt candidates.

**Versus Textgrad.** `Textgrad` relies on hand-crafted text losses and gradient-style backpropagation over natural language. In contrast, our method is entirely *gradient-free*: it requires no differentiable loss functions or template engineering. This avoids sensitivity to wording in loss templates and reduces dependence on diagnostic outputs, where weak language models often fail to generate meaningful diagnostic responses.

**Versus MIPRO.** `MIPRO` frames optimization as Bayesian search over (prompt, few-shot) pairs, requiring many trials and frequent evaluation metric calls. Its effectiveness hinges on having a well-defined evaluation metric, which is difficult to obtain in text-based games where no concise supervision signal exists. As a result, `MIPRO` consumes many tokens without achieving strong performance. Our method, by contrast, does not rely on explicit evaluation metrics. It can leverage diverse signals from self-play trajectories, achieving better performance with fewer model calls and without heavy trial scheduling.

**Versus GEPA.** `GEPA` extends `MIPRO`'s evaluation process by augmenting it with verbose textual feedback and repeatedly querying an evaluation oracle until its call budget is exhausted, making it heavily dependent on the quality of the evaluation metric. Its key mechanism is a Pareto-based selection strategy, which identifies promising prompts from the candidate pool based on the Pareto frontier. However, the construction of this frontier relies strongly on the evaluation scores, and when the metric is not well-defined, the selected prompts may not be optimal. In contrast, our method replaces such reliance on external feedback with *experience-guided edits* distilled directly from self-play outcomes, while maintaining diversity through crossover and randomization. This design reduces token usage, improves robustness under noisy feedback, and removes dependence on external evaluation metrics.

## G FULL RESULTS

Table 10: Performance of the COPER method across three independent trials using GPT-4o-mini and Qwen2.5-7B-Instruct. Results are reported as mean win rates with corresponding standard deviations.

| COPER | Negotiation | | Imperfect Info | | Perfect Info |
|---|---|---|---|---|---|
| | SimpleNegotiation | TwoDollar | KuhnPoker | Briscola | Simpletak |
| **GPT-4o-mini** | | | | | |
| Trial 1 | 57.3% | 46.0% | 54.0% | 54.0% | 45.3% |
| Trial 2 | 55.3% | 62.7% | 57.3% | 38.0% | 40.7% |
| Trial 3 | 52.0% | 48.7% | 55.3% | 36.0% | 39.3% |
| Avg. | 54.9% | 52.4% | 55.6% | 42.7% | 41.8% |
| Std. | 2.69 | 8.95 | 1.68 | 9.87 | 3.15 |
| **Qwen2.5-7B-Instruct** | | | | | |
| Trial 1 | 48.0% | 53.3% | 60.7% | 39.3% | 37.3% |
| Trial 2 | 47.3% | 54.0% | 59.3% | 26.7% | 32.0% |
| Trial 3 | 48.7% | 38.0% | 60.0% | 27.3% | 32.7% |
| Avg. | 48.0% | 48.4% | 60.0% | 31.1% | 34.0% |
| Std. | 0.67 | 9.05 | 0.67 | 7.13 | 2.90 |

## H    GAME ENVIRONMENTS

These are the more detailed descriptions of the games we selected the following set of text-based games from `TextArena` (Guertler et al., 2025a) and `SPIN-Bench` (Yao et al., 2025).

**Simple Negotiation** (Nash, 1950) requires players to reason about trade-offs through the exchange of resources such as wood, wheat, sheep, brick, and ore. Each player aims to maximize the value of their inventory by making offers and counteroffers with their opponent. Success depends on the each player's ability to infer the opponent's valuation of resources and strategically increase their own portfolio without making disadvantageous trades.

**Two Dollar Game** (Rowe, 2001) is a classroom negotiation game where two players have to agree on how to divide a fixed sum of $2.00. Typically, players each receive private role instructions that impose certain constraints or encourage specific negotiation styles. This asymmetric information requires players to balance their objectives with compromises while inferring the opponent's position.

**Kuhn Poker** (Kuhn, 1951) is a simplified form of poker played with three cards (Jack, Queen, and King). Two players each receive one card, while the third remains unseen. A single round of betting follows, where players can check, bet, call, or fold. If neither folds, the winner is determined by the higher card.

**Briscola** (McLeod, 2023) is a traditional Italian trick-taking card game played with a 40-card deck. At the start, a single card is revealed to determine the trump suit, and each player is dealt a hand of cards. Players take turns playing one card per trick, with the highest card of the leading suit or the highest trump winning the round. The objective is to accumulate points by capturing valuable cards, requiring players to balance tactical play with long-term strategy and inference of the opponent's hand.

**Simple Tak** (Rothfuss, 2011) is a two-player connection game inspired by the traditional game Tak. Players place tiles on a square grid with the objective of forming a continuous path that connects opposite sides of the board. Unlike full Tak, stacking pieces is not allowed, though players may block their opponent's path by occupying critical spaces. The game emphasizes spatial reasoning, foresight, and the balance between advancing one's own path and disrupting the opponent's progress.

## I    INSIGHT CASE ANALYSIS

We categorize the recurrent failure modes observed across games into the following three classes:

1. **Feedback framing errors:** the agent occasionally frames an insight as advice specific to Player 1, when it should instead be articulated as general strategic guidance applicable to both players.

2. **Redundant insights:** the experience bank sometimes contains multiple insights that convey essentially the same underlying idea, leading to unnecessary repetition.

3. **Over Simplified strategies:** in certain cases, the agent proposes strategies that are locally reasonable but globally suboptimal.

> **Two Dollars Experience Bank Example**
>
> High-Quality Insight:
> "To improve future negotiations, I could clearly convey the importance of reaching an agreement within the limited rounds available, perhaps framing my offer as a time-sensitive opportunity that other player wouldn't want to miss, thereby encouraging a quicker consensus."
> Low-Quality Insight:
> "In future negotiations, Player 1 should ensure they respond promptly and thoughtfully to proposals to maintain control of the negotiation dynamics. Proactively making a counter-offer could help anchor the negotiation towards a more favorable outcome."

Figure 17: Two-Dollar analysis. The high-quality insight proposes a dynamic negotiation strategy for the agent. In contrast, the mistake insight is not generalizable and falls into a feedback-framing error. It presents the advice as "Player 1 should follow the rules," rather than offering guidance applicable to any agent.

> **Simple Negotiation Experience Bank Example**
>
> High-Quality Insight:
> "To improve negotiation outcomes, Player should analyze the resource preferences of other Player more closely and tailor offers to match those preferences, possibly by proposing trades that highlight the mutual benefits rather than assuming equal value among resources."
> Low-Quality Insight:
> "Ensure to always double-check the format of trade offers before submission to avoid mistakes that can lead to missed trading opportunities."
> " Before making any trade offers, Player should review the syntax for offers to ensure compliance with the game's established format and thereby avoid disqualification of trade attempts."
> " Before making an offer in future rounds, Player 0 should double-check their resource inventory to ensure they are only proposing items they actually have and adhere to the correct format for trade proposals."

Figure 18: Simple Negotiation analysis. The high-quality insight identifies a concept not explicitly stated in the game description: players are assigned different values to the same resource. In contrast, the mistake insights are each individually correct, but when viewed together, they become highly repetitive and provide little additional value.

> **Khun Poker Experience Bank Example**
>
> High-Quality Insight:
> "In future games, consider a strategy where you bet or call more frequently in early rounds, even with weaker cards, to increase potential pots and apply pressure on the opponent, especially when no initial bets are made"
> Low-Quality Insight:
> "In future games, always bet with K since it is the strongest card, and fold when holding Q or J."

Figure 19: Khun Poker analysis. The high-quality insight introduces a general strategic principle that applies across many situations, using early aggression to build the pot and pressure the opponent. The low-quality insight is overly rigid and oversimplifies optimal play, since "always bet with K " and "fold when holding Q or J" can make the agent predictable and fail in situations where checking would be more profitable.

Briscola Experience Bank Example

High-Quality Insight:
"In future turns, prioritize using the Ace or trump cards at key moments to control the trick."
Low-Quality Insight:
"To prevent format errors in future moves, Player 0 should practice submitting actions using the correct structure, or consider writing a reminder of the required format before beginning the game. This will help reinforce the correct action submission and enhance overall gameplay effectiveness.",
"Always double-check the move format before submitting an action in order to prevent invalid moves and maintain the flow of the game.",
"To avoid format errors in future games, Player 0 should practice inputting actions in the specified format before starting a match or utilize a checklist or reminder to reinforce the correct action structure during gameplay."

Figure 20: Briscola analysis. High-quality insights articulate transferable strategic principles that apply across many game states and meaningfully improve the agent's decision-making. In contrast, low-quality insights consist largely of repetitive reminders about action formatting.

## J   PROMPT CASE ANALYSIS

Simple Negotiation System Prompt Transfer to Simple Tak Example

Updated System Prompt:
" Evaluate Opponent current inventory and preferences before crafting an initial trade proposal. Offer a carefully curated mix of both lower-value and higher-value items that not only maximize your immediate gain but also align with Player 1's needs, enhancing the perceived mutual benefits of the exchange. Ensure your strategy incorporates diversified resources to attract Opponent interest, leveraging aggressive initial offers that target their most valued items while securing optimal benefits for yourself"
Sample Action:
"To maximize my chances of winning while also considering the current state of the board, I will place my stone in cell 6. This move not only blocks Player 0's potential path but also positions me closer to forming my own continuous path., my move is: (boxed[6])"
"To maximize my chances of forming a continuous path while also considering the opponent's move, I will place my stone in cell 0. This move not only allows me to start building a path from the top edge but also blocks potential paths for Player 0.will submit my move as follows: boxed[0]"
"The board is 4x4, and I have two adjacent O's in the top row (positions 4 and 5). The opponent has placed X's in 6 and 7, blocking the top row to the right. To build towards a left-right connection, I should extend my horizontal line leftward by placing in position 0, which is empty and adjacent to my O in 4. boxed[0]"

Figure 21: Simple Negotiation Prompt Transfer to Simple Tak. Although the updated prompt was not trained on Simple Tak, it encourages the model to explicitly reason from the opponent's perspective during its thought process, resulting in more consistent and reliable performance compared to the basic starting prompt.

