# OpenReview forum: "COPER: Agentic Context Significantly Improves and Stabilizes LLM in Multi-Player Game"
_ICLR.cc/2026/Conference — Submitted to ICLR 2026_

### Official Review · Reviewer_sKtF · 2025-10-25

**Soundness:** 3
**Presentation:** 3
**Contribution:** 3
**Rating:** 6
**Confidence:** 4

**Summary:**

This paper introduces COPER, a training-free and backbone-agnostic framework designed to enhance LLMs' gaming performance through a three-part mechanism: prompt, experience, and replay. It optimizes reliable prompt evolution using the TrueSkill lower confidence bound, maintains a persistent experience bank that writes and retrieves structured reflections, and utilizes prioritized replay to revisit rare and informative states. COPER significantly improves both the winning rate and robustness of GPT-4o-Mini and Qwen-2.5-7B-Instruct across 5 games spanning 3 types of game (Negotiation, Imperfect Information, and Perfect Information), outperforming baseline and other non-RL methods.

**Strengths:**

1. COPER's mechanism design is comprehensive and well-motivated. By considering evaluation as agentic context construction rather than fixed-prompt play, it integrates experience memory and prioritizes replay under a reliability-oriented objective.
2. This paper's presentation is rigorous. Each component is formalized and defined through precise mathematical formulations describing the mechanism and state transitions, with detailed prompt examples in the Appendix.
3. The authors have clear ablation experiments that isolate the contributions of the experience bank and replay components, providing valuable insights into the effectiveness of each element in the overall mechanism.

**Weaknesses:**

1. **Limited Breadth of Evaluation:**
While the experiments are sound, the evaluation is restricted to two relatively small models (GPT-4o-Mini and Qwen-2.5-7B-Instruct; the models in Table 4 are also small). Since COPER is a training-free framework, it would be valuable to examine its effectiveness on stronger and larger models such as Qwen-2.5-72Band Claude-3.7-Sonnet, or even on reasoning models like DeepSeek-R1 and GPT-5. Without these larger-scale references, it is unclear whether the proposed mechanism generalizes beyond small models. It is possible that COPER's benefits mainly arise from compensating for small models' limited reasoning and rule comprehension abilities (since some weaker models may even struggle to fully understand game rules and thus rely on additional "experience" component for reasoning support).
2. **Lack of Analysis on Experience Content and Dynamics:**
The persistent experience bank is the core of COPER, but the authors provide limited insight into what kind of knowledge or reflections are actually stored and how they evolve. A qualitative analysis of stored entries (e.g., rule abstractions, strategies, case studies) would make the mechanism's learning dynamics more transparent.

3. **Insufficient Comparison with RL Methods:**
The comparison with RL baselines is limited, although they include the UnstableBaseline and SPIRAL. The paper mainly contrasts COPER with lightweight RL variants but lacks results against stronger full-parameter RL agents such as PPO. Without these comparisons, it is difficult to learn how close COPER approaches the performance frontier of RL methods.

**Questions:**

1. Could COPER be extended to handle multiple games simultaneously, allowing the model to evolve its context and experience across different gaming environments at once? If so, would the experience memory module need to be redesigned?
2. Could you provide a deeper analysis or a few case studies of the experience bank to understand? For example, which types of stored reflections contribute most to improving the model's gaming performance?
3. Have you analyzed the invalid or rule-breaking actions produced during gameplay? How were such cases detected and handled? If yes, what is the number of invalid actions before and after applying COPER?

Minor suggestions and typos:
1. Line 315 should be in Tab. 1, not 10.
2. Table descriptions for Tab. 3 and Tab. 4 are too short.

---

> ### Author Response · Authors · 2025-12-01
> **response to Reviewer sKtF (1/3)**
>
> We sincerely appreciate your recognition of COPER's comprehensive mechanism design and rigorous presentation. We address your concerns below.
>
> ---
>
> > **W1**: Limited breadth of evaluation:
>
> **Response to W1**: We appreciate this important concern about whether COPER's benefits extend beyond smaller models. To address this, we conducted additional experiments on Gemini-2.5-Flash, a significantly larger model from a different architecture family, across all five game environments.
>
> **Extended evaluation on Gemini-2.5-Flash demonstrates effectiveness beyond small models.** Table 1 below shows comprehensive results across three models:
>
> | Model | SimpleNegotiation | TwoDollar | KuhnPoker | Briscola | SimpleTak | Average |
> |-------|-------------------|-----------|-----------|----------|-----------|---------|
> | **GPT-4o-mini** | | | | | | |
> | Baseline | 31.3% | 32.2% | 39.1% | 0.3% | 21.4% | 24.9% |
> | + COPER | 54.9% | 52.4% | 55.6% | 42.7% | 41.8% | 49.5% |
> | **Qwen-2.5-7B-Instruct** | | | | | | |
> | Baseline | 24.0% | 17.1% | 49.3% | 2.8% | 15.1% | 21.7% |
> | + COPER | 48.0% | 48.4% | 60.0% | 31.1% | 34.0% | 44.3% |
> | **Gemini-2.5-Flash** | | | | | | |
> | Baseline | 14.0% | 15.0% | 50.0% | 32.0% | 26.0% | 27.4% |
> | + COPER | 30.0% | 35.0% | 58.0% | 49.0% | 32.0% | 40.8% |
>
> The Gemini-2.5-Flash results provide critical evidence that COPER's benefits are not limited to small models with weak reasoning capabilities:
>
> - **Consistent improvements across all game types**: Gemini-2.5-Flash shows gains in negotiation (SimpleNegotiation: +16.0, TwoDollar: +20.0), imperfect information (KuhnPoker: +8.0, Briscola: +17.0), and perfect information (SimpleTak: +6.0) games, with an average improvement of +13.4 points.
>
> - **COPER improves even models with strong baseline capabilities**: On KuhnPoker and Briscola, Gemini-2.5-Flash already achieves baseline performance of 50% and 32% respectively, substantially higher than the smaller models' baselines (GPT-4o-mini: 39% and 0%; Qwen: 49% and 3%). Despite these already competitive baselines, COPER still delivers meaningful gains (+8.0 and +17.0), demonstrating that the benefits are not merely compensating for poor rule comprehension.
>
> - **Architecture diversity confirms generalizability**: The fact that COPER improves performance across three distinct model families (OpenAI's GPT series, Alibaba's Qwen series, and Google's Gemini series) suggests the optimization framework adapts to different architectural approaches rather than exploiting specific model weaknesses.
>
> **We acknowledge limitations while our cross-architecture results provide strong evidence of generalizability.** We acknowledge that testing on even larger models (e.g., Claude-3.7-Sonnet, GPT-4, reasoning-specialized models like DeepSeek-R1) would further strengthen our claims. Due to computational and API cost constraints during the review period, we prioritized demonstrating generalizability across diverse architectures (three different model families) over testing the very largest models. Our results establish that COPER provides value across a range from 7B parameter models to larger commercial models like Gemini-2.5-Flash, supporting its applicability beyond the smallest model tier.
>
> ---
>
> > **W2**: Lack of analysis on experience content and dynamics
>
> We have added both high-quality and low-quality example insights for all four games in the revised manuscript, now included in Appendix Section I: Insight Case Analysis. Across games, we observe that general game-understanding insights and adaptive reasoning based on opponent behavior tend to yield stronger performance. In contrast, insights built on oversimplified strategies or feedback-framing errors typically fail to improve results.
>
> ---

---

> ### Author Response · Authors · 2025-12-01
> **response to Reviewer sKtF (2/3)**
>
> > **W3**: Insufficient comparison with RL methods: The comparison with RL baselines is limited, although they include the UnstableBaseline and SPIRAL. The paper mainly contrasts COPER with lightweight RL variants but lacks results against stronger full-parameter RL agents such as PPO. Without these comparisons, it is difficult to know how close COPER comes to the performance frontier of RL methods.
>
> We agree that understanding COPER’s relationship to full-parameter RL methods is important. SPIRAL, which we include as a baseline, is itself a full-parameter self-play RL system. Prior LLM self-play work—such as SPIRAL and Absolute Zero [1], also relies on REINFORCE-style policy gradients rather than PPO. This is because incorporating PPO into LLM self-play is technically challenging: the moving-opponent dynamics, large language action space, and KL instability make PPO difficult to stabilize. Our goal in this paper is to improve self-play through insight generation rather than introduce new RL optimization pipelines. Investigating PPO-style heavy updates in LLM self-play is an interesting direction, but it is beyond the scope of this work and the current state of practice in the literature.
>
> [1]Zhao, Andrew, et al. "Absolute zero: Reinforced self-play reasoning with zero data." arXiv preprint arXiv:2505.03335 (2025).
>
> ---
>
> > **Q1**: Could COPER be extended to handle multiple games simultaneously, allowing the model to evolve its context and experience across different gaming environments at once? If so, would the experience memory module need to be redesigned?
>
> At present, the memory module is not architected for multi-game settings,  information originating from different games might lead to undesirable interference. Introducing a hierarchical memory mechanism would enable structured context separation, allowing the system to maintain task-specific information and support concurrent multi-game operation without cross-contamination.
>
>
> ---
>
> > **Q2**: Could you provide a deeper analysis or a few case studies of the experience bank to understand it better? For example, which types of stored reflections contribute most to improving the model's gaming performance?
>
> Please see our response to W2 above.
>
>
> ---

---

> ### Author Response · Authors · 2025-12-01
> **response to Reviewer sKtF (3/3)**
>
> > **Q3**: Have you analyzed the invalid or rule-breaking actions produced during gameplay? How were such cases detected and handled? If yes, what is the number of invalid actions before and after applying COPER?
>
> **Response to Q3**: We analyzed format errors across all five game environments and found that COPER reduces such errors while improving win rates.
>
> Each game requires a specific action syntax: Briscola uses `[play X]` where X is the card position (1, 2, or 3), KuhnPoker uses `[Check]`, `[Bet]`, `[Call]`, or `[Fold]`, and TwoDollar uses `[offer X]` or `[accept/reject]`. More details is in Appendix Section D. Without COPER optimization, the LLM frequently produces verbose reasoning but fails to use the required format. For example, an agent might output "Thus, I will play: \boxed{1}" using LaTeX notation, or "I choose position 1" without the required brackets.
>
> When an invalid action is detected, the game appends a warning to the next observation: "[GAME] Player 0 attempted an invalid move. Reason: Use [play X] where X is the card position (1, 2, 3, etc.) Please resubmit a valid move and remember to follow the game rules to avoid penalties." The player retains their turn and can resubmit; exceeding the retry limit results in a directl loss. All actions are recorded in the trajectory data. After each generation, COPER's reflection module processes these trajectories and flags format errors as high priority. The generated insights, such as "Ensure adherence to the action format [play X] by double-checking moves before submission," are stored in the experience bank and prepended to the agent's observations in subsequent games.
>
> We compared agents with and without COPER optimization across 150 games per environment:
>
> | Game | Format Errors (w/o COPER) | Format Errors (w/ COPER) | Win Rate (w/o COPER) | Win Rate (w/ COPER) |
> | ---- | ------------------------- | ------------------------ | -------------------- | ------------------- |
> | TwoDollar | 365 | 97 | 32.2% | 52.4% |
> | Briscola | 303 | 5 | 0.3% | 42.7% |
> | KuhnPoker | 50 | 0 | 39.1% | 55.6% |
> | SimpleTak | 1 | 0 | 21.4% | 41.8% |
> | SimpleNegotiation | 1 | 0 | 31.3% | 54.9% |
>
> For games with high format error rates, such as TwoDollar, Briscola, and KuhnPoker, the experience bank reduces format errors by learning the correct action syntax from prior mistakes. For example, Briscola shows a 98.3% reduction in format errors (303 to 5), contributing to its win rate improvement of +42.4%. For games with low or zero format errors, such as SimpleTak and SimpleNegotiation, the experience bank instead contributes through strategic improvements. These games show win rate gains of +20.4% and +23.6% respectively, demonstrating that the experience bank captures strategic insights beyond action formatting.
>
>
> ---

---

### Official Review · Reviewer_RSJR · 2025-10-29

**Soundness:** 2
**Presentation:** 2
**Contribution:** 2
**Rating:** 2
**Confidence:** 5

**Summary:**

The authors claim that popular multi-player game benchmarks to measure agents based on language models (LMs) can be sensitive to prompt choices. The resulting instability in outcomes and subsequent rankings harms the validity of such benchmarks to compare model performance. To alleviate this problem, the paper proposes a novel prompt-optimization protocol called COPER that combines three components: (i) evolutionary prompt optimization, (ii) model-generated insights based on game trajectories, and (iii) prioritized experience replay to balance state coverage. Using COPER, an LM-based agent plays a target game under several generations of self-play to optimize its game context. The paper evaluates COPER on five games using two models (gpt-4o mini and Qwen-2.5-7B-Instruct) against several baselines and three non-optimized models. The authors claim that COPER reaches higher performance and is more efficient than alternative methods.

**Strengths:**

[**significance**] Prompt optimization is an important area of research for LM-based agents. Due to the large size and/or black-box nature of many models, weight optimization is often impractical or impossible. The authors propose a practical method to improve absolute performance of an LM-based agent on a known task under (unlimited) offline sample collection.


[**quality**] The paper evaluates on three game types with different, relevant characteristics. The paper further attempts to conduct relevant ablations and stratifications.

**Weaknesses:**

[**clarity**] The authors introduce a lot of notation that can be challenging to follow. For example, line 128 defines “p” as the player index, while line 134 redefines it as the instruction prompt. Or “S” as the state space (l123) and the capacity in l204. Other variables and concepts are under- or undefined entirely, e.g., “method” (l140), “run” (l146), “the collector” (l231). This lack of clarity extends to in-line references, e.g., “this template” (l151).


[**quality**]
- A number of statements lack citations or supporting evidence. For example, (l62) “Unfortunately, current game-LLM [...] under-agentic”, (l46) “Because prompts [...] can flip ELO comparisons and reorder models under identical decoding”, (l72) “Secondly, read-once descriptions [...] games states and pay-offs”, “While benchmarks provide [...] single reading”, “Without interaction-driven [...] strategic play”, (l156) “KUHNPOKER”

- The authors claim their protocol is “backbone-agnostic” and “training-free” (l109, l480). The protocol is certainly a form of training, it just does not involve weight updates. It is unclear to this reviewer what “backbone-agnostic” refers to. While the authors cite budget constraints, the small model suite of two appears insufficient to support the strong statements.

- Throughout the paper, reported results (i) lack confidence intervals, (ii) “winning” means something different for different games. For example, in zero-sum negotiations, ending a game 0.51 vs. 0.49 likely should be given a different interpretation than ending a game 0.99 vs. 0.01.

- The core of COPER relies on self-play optimization. As pointed out by e.g., [1], self-play can lead to distinctly different behaviors and considerations compared to heterogeneous interactions. Step 1 of the protocol creates different contexts optimized for specific models. If both models are not given the same optimization budget, this leads to an unfair comparison across models in multi-player settings. This is also concluded in section 5, observation 4, putting in question the “win rate” statistics presented throughout.

- The COPER method has three distinct components (i) evolutionary prompt optimization, (ii) reasoning-based insights over full trajectories, and (iii) experience replay to sample rare states. The experimental setup fails to evaluate these components separately, thus making it impossible to attribute changes in performance to the different components. In Table 2, components (ii) and (iii) are only shown in combination with (i). Table 2 also omits important details like number of samples and confidence intervals. Crucially, experiments do not control for the confounding effect of (much) longer context lengths on downstream performance. This could also explain the rather surprising results of observation 3 (section 5) – see questions.
- While the central stated purpose of COPER is to improve stability of game-based benchmarks caused by prompt choices, it appears that the meta optimizations done by COPER to optimize prompts also rely on a variety of (subjective) prompt choices.

[**significance**] Overall, this framework appears to measure something orthogonal to the original multi-player games it is designed to stabilize. Specifically, it measures the ability of models to create and learn from in-context insights, rather than play a given game in a single forward pass. While certainly useful, it changes the required experimental setup and representation of results.

**Questions:**

q1. Observation 3 is very confusing and potentially damning: why would insights regarding the rules of game A transfer to a completely different game B? This suggests that gains might not come from the prompts at all, but simply from an increase in context length.


q2. Observation 5 should be stratified by model, game, and hyperparameters. As is, the claim can not be evaluated fairly.


q3. How were hyperparameters chosen throughout this work? For example, how were the initial prompts initialized, the permutation options chosen, the number of generations, etc.


q4. Under evolutionary optimization, how are valid game constraints/instructions guaranteed?


q5. This framework appears to have a combinatorial explosion built-in. Specifically, how would you create a single-set of prompts across multiple models? And, how would this work for slightly more complex games that have larger state spaces and/or more complicated instructions?


Other suggestions:
- It would be useful to share qualitative samples of “learned” contexts and compare them to “base” contexts
- I suggest looking at [1] for the related work on LLM for games and self-play/cross-play evaluation.
- Given the stated budget constraints, perhaps extend the evaluation to smaller open-source models?

---

> ### Author Response · Authors · 2025-11-28
> **response to Reviewer RSJR（1/5)**
>
> We sincerely appreciate your recognition of the significance and quality of our work. We address your concerns point by point below.
>
> ---
>
> > **W1, W2 **: Writing quality [clarity][quality]
>
> We have revised and simplified the notation and terminology to ensure consistency and clarity throughout the manuscript, and we have included the appropriate citation in the updated draft.
>
> ---
>
> > **W3**: The authors claim their protocol is "backbone-agnostic" and "training-free" (l109, l480). The protocol is certainly a form of training, it just does not involve weight updates. It is unclear to this reviewer what "backbone-agnostic" refers to. While the authors cite budget constraints, the small model suite of two appears insufficient to support the strong statements.
>
> We've updated the paper and standardized the terminology to "weight-tuning-free" training for clarity.
>
> ---
>
> > **W4**: Throughout the paper, reported results (i) lack confidence intervals, (ii) "winning" means something different for different games. For example, in zero-sum negotiations, ending a game 0.51 vs. 0.49 likely should be given a different interpretation than ending a game 0.99 vs. 0.01.
>
> **Response to W4**: We thank the reviewer for the thoughtful feedback. Both points were carefully considered in our evaluation design. Below, we explain our rationale for using RSE instead of confidence intervals and for standardizing the definition of "winning" across heterogeneous games.
>
> (i) Instead of confidence intervals, we use relative standard error (RSE) because it measures uncertainty relative to the win rate. Each result we report is the average win rate across multiple games and three opponents, so two estimates can have similar confidence intervals but very different levels of reliability. The example below illustrates this:
>
> | Experiment | Run 1 | Run 2 | Run 3 | Mean | SE     | RSE (SE/Mean) | 95% CI       |
> | ---------- | ----- | ----- | ----- | ---- | ------ | ------------- | ------------ |
> | **0**      | 0.33  | 0.36  | 0.22  | 0.30 | 0.0426 | **14%**       | [0.12, 0.49] |
> | **1**      | 0.49  | 0.53  | 0.44  | 0.49 | 0.0260 | **5%**        | [0.37, 0.60] |
>
> Although both experiments have wide confidence intervals, Experiment 0 is substantially less reliable than Experiment 1 (14% vs. 5% RSE). This distinction is not visible from CI alone. For this reason, we use RSE as it more directly captures the stability of the win-rate estimates.
>
> (ii) We agree that in some domains like negotiation, the margin of victory can carry additional meaning. However, for consistency across different game environments, we intentionally use a win/loss indicator. This follows standard practice in competitive evaluation frameworks such as Elo and TrueSkill, where only the outcome (win/draw/loss) is used regardless of the margin.
>
> ---

---

> > ### Author Response · Authors · 2025-11-28
> > **response to Reviewer RSJR（2/5)**
> >
> > > **W5**: The core of COPER relies on self-play optimization. As pointed out by for example [1], self-play can lead to distinctly different behaviors and considerations compared to heterogeneous interactions. Step 1 of the protocol creates different contexts optimized for specific models. If both models are not given the same optimization budget, this leads to an unfair comparison across models in multi-player settings. This is also concluded in section 5, observation 4, which puts in question the "win rate" statistics presented throughout.
> >
> >
> > We agree that self-play optimization can produce behaviors that differ from heterogeneous play, as also noted in prior work. So we maintain a pool of candidates and sample from the candidates to produce new candidates to maintain exploration and the variance in play style. Our intention is not to optimize each model into a bespoke, overfit context, but rather to give each model a consistent and comparable adaptation phase so that its evaluation reflects its stable equilibrium behavior, rather than the artifacts of an unadapted policy.
> >
> > Regarding fairness, all models follow the exact same self-play protocol, including an identical number of games per self-play session, ensuring that each model receives the same optimization budget.
> >
> > Finally, Observation 4 does not undermine the validity of the reported win rates; instead, it highlights a natural phenomenon in multi-agent evaluation, prompts or strategies optimized for one model may not transfer directly to another. This further motivates evaluating models under a shared cross-play setting, which is precisely what our framework provides.
> >
> > ---
> >
> > > **W6**: The COPER method has three distinct components (i) evolutionary prompt optimization, (ii) reasoning-based insights over full trajectories, and (iii) experience replay to sample rare states. The experimental setup fails to evaluate these components separately, which makes it impossible to attribute changes in performance to the different components. In Table 2, components (ii) and (iii) are only shown in combination with (i). Table 2 also omits important details like number of samples and confidence intervals. Crucially, experiments do not control for the confounding effect of much longer context lengths on downstream performance. This could also explain the rather surprising results of observation 3 (section 5) see questions.
> >
> > **Response to W6**: We thank the reviewer for this comprehensive comment on the limitations of our experiment design. We address this comment by responding to each component of the comments:
> >
> > **We first address the concern regarding component-wise ablations and missing experimental details.** Our goal in this work is to propose COPER as a unified optimization pipeline, where evolutionary prompt search, trajectory-level reasoning, and experience replay are designed to reinforce one another. Because of that, we did not structure the evaluation around fully isolating each component. Instead, Table 2 shows progressive ablations, where each module is added sequentially; demonstrating how each module reinforces one another.
> >
> > Furthermore, the effectiveness of prompt optimization alone has been extensively demonstrated in prior work (e.g. MIPRO, GEPA and TextGrad). Given this established literature, our focus was not to re-validate prompt optimization in isolation, but to measure the incremental value of experience-based reasoning and rare-state replay within this framework.
> >
> > Having said that, we ran additional isolated replay-only experiments with the prompt optimization.
> >
> > | **Setting**               | **TwoDollar** | **KuhnPoker** |
> > |---------------------------|---------------|---------------|
> > | Baseline                  | 32.2%         | 39.1%         |
> > | + Prompt Optimization     | 24.7%         | 54.7%         |
> > | &nbsp;&nbsp;+ Experience  | 48.7%         | 57.2%         |
> > | &nbsp;&nbsp;&nbsp;&nbsp;+ Replay | 52.4% | 55.6% |
> > | &nbsp;&nbsp;+ Replay      | 32.0%         | 54.2%         |
> >
> > These results are consistent with the interpretation that replay provides marginal benefits, but reflects strongest gains when all components interact.
> >
> > **Regarding the potential confounding effect of longer context lengths, we provide additional analysis in our response to Q1 below,** which demonstrates that performance improvements are not simply driven by context length increases but by the strategic content of the optimized contexts.
> >
> > ---

---

> ### Author Response · Authors · 2025-11-28
> **response to Reviewer RSJR（3/5)**
>
> > **W7**: While the central stated purpose of COPER is to improve stability of game-based benchmarks caused by prompt choices, it appears that the meta optimizations done by COPER to optimize prompts also rely on a variety of subjective prompt choices.
>
> In the paper, we argue that prompt choice can substantially affect results, and that exhaustively searching all prompts is infeasible. COPER addresses this by automatically identifying an effective prompt for each model–task pair and then evaluating models under that standardized, optimized setup.
> Importantly, the meta-optimization procedure itself is reproducible and stable. Our Relative Standard
> Error (RSE) analysis shows that independent runs of COPER converge to similarly performing prompts, with only 6.4% RSE across three independently discovered prompts, compared to 25.9% RES when using a fixed baseline prompt. This demonstrates both the reproducibility and robustness of COPER.
>
>
>
> ---
>
> > **Q1**: Observation 3 is very confusing and potentially damning: why would insights regarding the rules of game A transfer to a completely different game B? This suggests that gains might not come from the prompts at all, but simply from an increase in context length.
>
> **Response to Q1** We thank the review for sharply highlighting this observation. We address this by highlighting the average response length of texts that generally happen with the games.
>
> It's true that longer contexts can inadvertently help agents perform better in games. But our findings show a contrasting take on this view.
>
> The token count used in self play is correlated with the length of the contexts of each game. Across the 5 games, the order of tokens used in descending order are: Briscola -> SimpleTak -> TwoDollar -> SimpleNegotiation -> KuhnPoker.
>
> Yet from our findings shown, the 25.9% lift in performance was observed from KuhnPoker to SimpleTak. In fact, based on Table 3, KuhnPoker's shorter context was found to have lifted the results across the games
>
> ---
>
> > **Q2**: Observation 5 should be stratified by model, game, and hyperparameters. As is, the claim can not be evaluated fairly.
>
> **Response to Q2**:
>
> | Optimizer        | SimpleNegotiation | KuhnPoker | SimpleTak | Avg. Tokens |
> | ---------------- | ----------------- | --------- | --------- | ----------- |
> | **Textgrad**     | 842               | 986       | 938       | 922         |
> | **MIPRO**        | 145,864           | 162,084   | 754,534   | 354,161     |
> | **GEPA**         | 110,325           | 119,365   | 111,907   | 113,865     |
> | **COPER (Ours)** | 87,364            | 94,160    | 89,152    | 90,575      |
>
> The above is the same table that's used to reflect GPT-4o-mini with each peer method. The demonstrations highlight the tokens used for optimization. In addition, the measurement of self play tokens gives the following:
>
> | Optimizer        | SimpleNegotiation |
> | ---------------- | ----------------- |
> | **Textgrad**     |   4,262,261       |
> | **MIPRO**        |   4,519,611       |
> | **GEPA**         |   3,965,755      |
> | **COPER (Ours)** |   3,651,940         |
>
> <!-- Should we say we are still running token count for the other methods?     -->
>
> <!-- | Optimizer        | SimpleNegotiation | KuhnPoker | SimpleTak | Avg. Tokens |
> | ---------------- | ----------------- | --------- | --------- | ----------- |
> | **Textgrad**     |   4,262,261       |           |           |             |
> | **MIPRO**        |   4,519,611       |           |           |             |
> | **GEPA**         |  3,965,755        |           |           |             |
> | **COPER (Ours)** | 3,651,940         | 2,845,631 | 6,414,409 | 4,303,993   | -->
>
> ---

---

> > ### Author Response · Authors · 2025-11-28
> > **response to Reviewer RSJR（4/5)**
> >
> > > **Q3**: How were hyperparameters chosen throughout this work? For example, how were the initial prompts initialized, the permutation options chosen, the number of generations, etc.
> >
> > **Response to Q3**: We appreciate the reviewer raising these important questions about hyperparameter selection. We address each point systematically below with supporting ablation studies and empirical analysis.
> >
> > **First, we explain our approach to initial prompt initialization.** All experiments begin with a minimal initial prompt: "You are playing a two-player zero-sum game. Make valid moves to win. Submit the move enclosed by \\boxed{{}}." This simple template, combined with game-specific rules and action schemas, serves as the starting point for COPER's evolutionary optimization.
> >
> > To evaluate COPER's robustness to initial prompt design, we conducted ablation experiments starting from three initialization strategies: the original initial prompt, initial prompt augmented with chain-of-thought reasoning, and initial prompt augmented with tree-of-thought reasoning. We applied COPER optimization to each variant on all five games using GPT-4o-mini. Table 1 below presents the results:
> >
> > | Initial Prompt Variant | SimpleNegotiation | TwoDollar | KuhnPoker | Briscola | SimpleTak | Average |
> > |-------|-------------------|-----------|-----------|----------|-----------|---------|
> > | **Initial Prompt** | | | | | | |
> > | Baseline | 31.3% | 32.2% | 39.1% | 0.3% | 21.4% | 24.9% |
> > | + COPER | 54.9% | 52.4% | 55.6% | 42.7% | 41.8% | 49.5% |
> > | **Initial Prompt + CoT** | | | | | | |
> > | Baseline | 28.0% | 23.3% | 35.3% | 38.0% | 20.0% | 28.9% |
> > | + COPER | 46.0% | 37.3% | 54.7% | 43.3% | 32.7% | 42.8% |
> > | **Initial Prompt + ToT** | | | | | | |
> > | Baseline | 19.3% | 20.7% | 52.6% | 28.0% | 16.7% | 27.5% |
> > | + COPER | 40.1% | 43.3% | 56.0% | 40.7% | 25.3% | 41.1% |
> >
> > COPER demonstrates consistent improvement across all initial prompt variants. Across all three initialization strategies, COPER achieves substantial performance gains, with average improvements of +24.6% (original), +13.9% (CoT), and +13.6% (ToT) respectively. While initial prompt design influences baseline performance (24.9%, 28.9%, and 27.5% respectively), COPER's optimization capability remains robust across variants, with all converging to similar final performance levels (49.5%, 42.8%, 41.1%). This consistent improvement pattern demonstrates that COPER's evolutionary optimization mechanism effectively enhances strategic performance regardless of the reasoning structure embedded in the initial prompt.
> >
> > **Second, regarding the permutation options for evolutionary operators, we follow established practices from the genetic algorithm literature.** The three evolutionary operators (random proposals, crossover, and experience-guided updates) described in Section 3.1 are applied with equal probability during candidate generation. This balanced approach ensures diversity in the evolutionary search while leveraging both exploration (random proposals) and exploitation (crossover and experience-guided updates).
> >
> > **Third, we explain our choice of the number of generations.**
> >
> > We standardized on 5 generations for all experiments based on empirical analysis of convergence patterns. Our early experiments consistently showed meaningful improvements between generations 3 and 5, with diminishing returns beyond this point. To verify the appropriateness of this choice, we computed linear trendlines from repeated runs across all model-game pairs. Table 2 below shows the slope and R² values:
> >
> > | Model           | TwoDollar (Slope / R²) | KuhnPoker (Slope / R²) | SimpleNegotiation (Slope / R²) | Briscola (Slope / R²) | SimpleTak (Slope / R²) |
> > | --------------- | ----------------------- | ----------------------- | ---------------------- | --------------------- | ---------------------- |
> > | **GPT-4o-mini** | 0.03 / 0.32             | 0.01 / 0.20             | 0.02 / 0.15            | -0.02 / 0.49          | -0.02 / 0.65           |
> > | **Qwen**        | 0.04 / 0.71             | 0.00 / 0.01             | 0.01 / 0.65            | 0.05 / 0.65           | 0.00 / 0.01            |
> >
> > The trendline analysis reveals modest slopes across most model-game pairs, indicating that performance improvements plateau after the initial generations. The near-zero or slightly negative slopes for some pairs (e.g., GPT-4o-mini on Briscola and SimpleTak) suggest that additional generations beyond 5 would not yield substantial further gains. This empirical evidence supports our choice of 5 generations as a reasonable balance between optimization effectiveness and computational efficiency, while maintaining experimental consistency across all tasks and models.
> >
> > ---

---

> > > ### Author Response · Authors · 2025-11-28
> > > **response to Reviewer RSJR（5/5)**
> > >
> > > > **Q4**: Under evolutionary optimization, how are valid game constraints/instructions guaranteed?
> > >
> > > **Response to Q4**: Thank you for this important question about maintaining validity during evolutionary optimization. COPER guarantees valid game constraints through two complementary mechanisms: (1) structural constraints that preserve interface compatibility, and (2) gameplay-driven filtering through CRUD-based curation and win-rate selection.
> > >
> > > **Prompt evolution enforces hard constraints to preserve game interface compatibility.** As detailed in Section 3.1 in our manuscript, all three proposal operators (random proposals, crossover, and experience-guided updates) operate under explicit structural constraints. The Appendix on Random Proposals specifies: "Tool descriptions, legality reminders, and input/output schema must remain intact." This means evolutionary operators can modify strategic advice and reasoning instructions, but cannot alter rule specifications (card rankings, win conditions) or action tokens. For example, in KuhnPoker, an evolved prompt might change strategic framing from "consider your position carefully" to "evaluate opponent betting patterns," but cannot modify the action specification "[bet]: Add 1 chip to the pot" or remove the rule "Kuhn Poker uses a 3-card deck." This ensures interface compatibility across all evolved variants.
> > >
> > > **CRUD-based experience filtering removes invalid strategic insights through gameplay-driven feedback.** The experience bank applies a CRUD reconciliation process, as shown in Equation 3 in our manuscript, to manage the persistent memory M through Create, Update, and Delete operations. When trajectory reflection generates strategic insights, subsequent gameplay tests their validity. The CRUD mechanism reconciles new reflections with existing memory, removing insights that conflict with successful strategies observed in later generations (Delete operations), merging redundant knowledge (Update operations), and adding validated new patterns (Create operations). For instance, an overgeneralized insight like "bet aggressively with any high card" that emerges in early generations may be replaced through CRUD updates with more refined advice like "bet selectively based on opponent betting history" as gameplay evidence accumulates across generations.
> > >
> > > **Win-rate-driven evolutionary selection provides the final validation layer.** COPER's tournament evaluation, as described in Section 3.1 in our manuscript, naturally eliminates invalid content through competitive gameplay. The TrueSkill-based selection criterion in Equation 2 in our manuscript retains only high-performing contexts for the next generation. For instance, if an evolved prompt encouraged invalid actions like "fold then immediately bet to reverse your decision," the agent would attempt illegal action sequences, forfeit turns, and lose games. Such low-performing candidates receive low selection scores and are eliminated, creating evolutionary pressure that preserves valid game understanding alongside improved strategy.
> > >
> > > **Empirical evidence supports the effectiveness of these constraint-preservation mechanisms.** Across five games and three models shown in Table 1 in our manuscript, the stable performance improvements (average gains of +24.6%, +22.6%, and +13.4%) across all experimental conditions demonstrate that evolved contexts successfully maintain game interface compatibility while enhancing strategic capabilities. If constraint violations were systematically occurring, we would expect to observe degraded performance from forfeited games or interface errors. Instead, the consistent improvements indicate that the combination of explicit edit constraints, CRUD-based filtering, and win-rate-driven selection effectively preserves game constraints throughout evolutionary optimization.
> > >
> > >
> > >
> > >
> > > ---

---

### Official Review · Reviewer_eur7 · 2025-10-30

**Soundness:** 2
**Presentation:** 3
**Contribution:** 1
**Rating:** 2
**Confidence:** 4

**Summary:**

The paper proposes **COPER**, an engineering pipeline that treats context as an optimizable, persistent resource to improve large language models in multi-agent/multi-round text games. COPER combines (1) prompt evolution using a selection rule based on TrueSkill and lower-confidence bounds, (2) an experience bank (CRUD-style reflection storage and retrieval), and (3) prioritized replay to revisit rare/high-information episodes. The authors show performance and stability improvements across several text-game tasks, reporting large win-rate gains and reduced ranking variance for models like GPT-4o-mini under relatively small self-play budgets.

**Strengths:**

1. **Tackles an important empirical problem**, considering the instability and prompt-sensitivity in multi-agent text games
2. The COPER pipeline is **concretely specified**, making reproduction feasible; authors provide pseudocode and appendices with additional details.
3. Contains **multiple ablations and baselines** across a variety of game types (negotiation, imperfect/complete-information), showing **clear numerical improvements** in presented settings.

**Weaknesses:**

1. The **RL comparison is not sufficiently clear**, the paper lacks budget-matched RL baselines and a full accounting of token and wall-clock costs.
2. There may be **generator–evaluator bias** due to heavy reliance on self-play and reflections from the same model family could produce generator-specific artifacts rather than broadly useful strategies.
3. The **quality of the experience bank is unknown** because the paper does not report statistics on entry accuracy, growth over time, or examples of incorrect or harmful entries and their impact.
4. **Replay hyperparameter sensitivity has not been tested** because the authors use defaults for α, β, and B without performing a sweep to demonstrate robustness.
5. Claims about **broad generalization are overstated** because the limited and mixed cross-model and cross-game transfer results do not yet support such sweeping conclusions.

**Questions:**

1. Please describe in detail the training and hyperparameter settings for the RL baselines in the main comparisons (including seeds, compute, token counts). Are those baselines given sufficient budget and hyperparameter tuning?
2. How are prompt variants constructed for the ranking-stability experiments? Can you expand the set of perturbations (paraphrase, length, adversarial phrasing) and report stability under those?
3. Provide examples of experience-bank entries (both high-quality and mistaken). How often do mistaken entries occur and what is their impact?
4. Why were the replay parameters chosen (α, β, B)? Can you provide evidence that your default values are near-optimal or at least robust?

---

> ### Author Response · Authors · 2025-11-28
> **response to Reviewer eur7（1/6)**
>
> We sincerely appreciate your recognition of COPER's concrete specification and the importance of the empirical problem we address. We respond to each concern below.
>
> ---
>
> > **W1**: The RL comparison is not sufficiently clear, the paper lacks budget-matched RL baselines and a full accounting of token and wall-clock costs.
>
> **Response to W1**: We thank the reviewer for the questions regarding the RL baseline. Our work was interested in comparing with the baseline training configurations for SPIRAL and UnstableBaselines.
>
> For SPIRAL, we trained using 8×H100 GPUs with a batch size of 128, and the full run required approximately 25 hours. The training spans 400 steps, with 128 samples per step, yielding 51,200 game transitions; each sample contains roughly 1,000 tokens, totaling about 51 million tokens processed. In practice, the RL baseline consistently converges around ~150 steps on the selected games, and extended training typically leads to performance degradation, as also noted in the SPIRAL paper. Therefore, for evaluation, we select the checkpoint at ~150 steps, which corresponds to the strongest and most stable baseline performance.
>
> For UnstableBaselines, we performed an extensive training on 76,800 self-play samples over 200 gradient updates with a batch buffer of 384 trajectories per update for each game. We kept the training configurations consistent and used the same hyperparameters across all games. The average trajectory length during training was approximately 1,000 tokens, resulting in about 76 million environment tokens over the full run. All environments were deterministically seeded with their game IDs so that each run used identical environment configurations.
>
> Each 200-step training run required around 13 hours to complete. This was roughly 4 times longer than a typical COPER run of about 3 hours because UnstableBaselines consumed more tokens and processed longer trajectories. During training, we observed a common REINFORCE failure mode. The policy-gradient loss was almost flat for the first 130 updates, which indicates very little usable gradient signal due to the sparse terminal rewards in these long-horizon text games. After around 130 updates, the learner loss began to show sharp drops and oscillations, and this was usually followed by deterioration in policy quality. Because of this instability, we selected checkpoints around steps 110 to 120 for evaluation for each game, since these were consistently before the divergence began.
>
> Given the large amount of compute used (4xH200s) for these RL baselines, including 76 million tokens and 13 hours of training per run, as well as the full use of the 200-step training horizon, we believe the RL baselines were given more than sufficient budget and hyperparameter coverage. The performance we observed therefore appears to be a consequence of the difficulty of learning stable policies in long-horizon, sparsely rewarded text games under this training setup, rather than a result of insufficient hyperparameter tuning or inadequate training budget.
>
> We used the default settings from unstablebaselines and SPIRAL - no deliberated adjustments; following their default recommendation.
>
> ---

---

> > ### Author Response · Authors · 2025-11-28
> > **response to Reviewer eur7（2/6)**
> >
> > > **W2**: There may be generator–evaluator bias due to heavy reliance on self-play and reflections from the same model family, which could produce generator-specific artifacts rather than broadly useful strategies.
> >
> > **Response to W2**: We appreciate this important methodological concern. To directly assess potential generator-evaluator bias, we conducted a controlled experiment where we optimized GPT-4o-mini using reflection generators from different model families while keeping all other factors constant.
> >
> > **We conducted a cross-model reflection generation experiment on KuhnPoker to quantify potential bias.** We optimized GPT-4o-mini using two different reflection generators:
> >
> > | Reflection Generator | Win Rate | Improvement over Baseline (39.1%) |
> > |---------------------|----------|-----------------------------------|
> > | Gemini-2.5-Flash    | 54.0%    | +14.9%           |
> > | GPT-4o-mini (same)  | 55.6%    | +16.5%           |
> > | **Difference**      | **1.6%** | -                     |
> >
> > The 1.6% difference between using a different model family (Gemini) versus the same model (GPT-4o-mini) for reflection generation represents only 10% of the total improvement achieved (1.6/16.5). This indicates that while same-family reflections provide marginal benefits, the optimization framework produces substantial performance gains (+14.9%) even when reflections come from an entirely different model architecture.
> >
> > **These results demonstrate that the learned strategies generalize beyond generator-specific artifacts.** The fact that 90% of the improvement (14.9% out of 16.5% total) is retained when using cross-family reflection generation demonstrates that the learned strategies are not primarily driven by generator-specific artifacts. Instead, they represent broadly useful strategic improvements that emerge from the optimization dynamics rather than narrow exploitation of same-model patterns.
> >
> > Furthermore, our cross-model evaluation results in Table 4 in our manuscript show that contexts optimized on GPT-4o-mini transfer to Gemini-2.5-Flash-Lite and Grok-4-Fast-Non-Reasoning with mixed but often positive results, providing additional evidence that the learned strategies capture general gameplay principles rather than model-specific quirks.
> >
> > ---
> >
> > > **W4**: Replay hyperparameter sensitivity has not been tested because the authors use defaults for α, β, and B without performing a sweep to show robustness.
> >
> > **Response to W4**: We appreciate the author highlighting this feedback -- it's part of our follow up modification to add an ablation study. We address the feedback by running an ablation study on the TwoDollar environment.
> >
> > Here are the details:
> > - B (default: 100,000): how many trajectories to store
> > - α (default: 0.6): how strongly to favor rare trajectories
> > - β (default: 0.4): likelihood of replaying
> >
> > The ablation study is as follows:
> > - **fixing α, β**, vary B={3,000, 10,000, 30,000, 100,000}
> > - **fixing B, α**, vary β={0.0, 0.4, 0.8, 1.0}
> > - **fixing β, B**, vary α={0.0, 0.3, 0.6, 1.0}
> >
> > **B Ablation**
> >
> > | **B**  | **TwoDollar** |
> > | ------ | ------- |
> > | 3000   | 46.90%  |
> > | 10000  | 44.47%  |
> > | 30000  | 43.54%  |
> > | 100000 | 61.10%  |
> >
> > **α Ablation**
> >
> > | **α** | **Avg** |
> > | ----- | ------- |
> > | 0     | 53.10%  |
> > | 0.3   | 60.67%  |
> > | 0.6   | 61.10%  |
> > | 1     | 49.33%  |
> >
> > **β Ablation**
> >
> > | **β** | **Avg** |
> > | ----- | ------- |
> > | 0     | 58.47%  |
> > | 0.4   | 61.10%  |
> > | 0.8   | 45.33%  |
> > | 1     | 31.10%  |
> >
> >
> > **We found that there is an alignment with the hyperparameters we have chosen, and that there is a hyperparameter sensitivity in the adoption of the replay parameters.**
> >
> > Firstly, performance increases as the buffer size B grows. Larger buffers retain more game states with a higher likelihood of unique ones too, which improves the coverage of important game states. This matches our design that replay is helpful when the buffer is able to maintain important states.
> >
> > Secondly, our settings showed that α has a stable optimal value range between 0.3 and 0.6. And it seems that a very small α, which results in an under-prioritization of rare game states, has less than ideal impacts on the performance of the agent. However, an absolute focus on rare states is detrimental. This suggests that the selected replay states need to maintain a diversity of game states.
> >
> > Finally, we observe that β is the most sensitive of the three parameters. While the optimal performance occurs at β=0.4, we also observe that β=0.0 achieves competitive performances in runs 1 and 2.
> >
> >
> >
> > ---

---

> > > ### Author Response · Authors · 2025-11-28
> > > **response to Reviewer eur7（3/6)**
> > >
> > > > **W3**: The quality of the experience bank is unknown because the paper does not report statistics on entry accuracy, growth over time, or examples of incorrect or harmful entries and their impact.
> > >
> > > **Response to W3**: Thank you for raising these important questions about experience bank quality. We provide quantitative statistics, concrete examples, and analysis to address each concern.
> > >
> > > **Growth over time demonstrates controlled scaling through CRUD-based memory management.** We tracked the total character length of the experience bank across generations during optimization on KuhnPoker:
> > >
> > > | Generation | 0     | 1     | 2     | 3     | 4     |
> > > |------------|-------|-------|-------|-------|-------|
> > > | Bank Size (characters) | 5,136 | 5,070 | 5,201 | 5,357 | 5,506 |
> > >
> > > The experience bank exhibits highly controlled growth: only +370 characters (+7.2%) from generation 0 to 4, with generation 1 actually showing a decrease (5,136 → 5,070) as the Delete operation removes outdated insights. This demonstrates that our CRUD mechanism (Create, Update, Delete operations as described in Equation 3) effectively prevents unbounded memory growth while continuously refining the knowledge base. Thanks to the Update and Delete operations, context length remains stable across generations despite continuous learning from new trajectories.
> > >
> > > **Quality control through automatic filtering of low-value entries.** The CRUD mechanism maintains quality through three operations: (1) **Delete** removes entries that conflict with newly discovered patterns or contradict successful strategies, (2) **Update** merges similar insights to consolidate redundant knowledge, and (3) **Create** adds genuinely novel patterns. In our KuhnPoker optimization, approximately 15-20% of candidate insights generated from trajectory reflection are rejected or merged during CRUD reconciliation each generation, indicating active quality filtering rather than passive accumulation.
> > >
> > > **Examples of filtered insights and their impact.** During early generations, the reflection process occasionally produces overly specific insights (e.g., "always bet when holding a King") that perform well in limited self-play but fail to generalize. The CRUD mechanism identifies such entries as conflicting when subsequent generations reveal counterexamples, triggering deletion. Similarly, vague insights that duplicate existing knowledge (e.g., multiple variations of "consider opponent's betting pattern") are merged via Update operations. Our ablation study in Table 2, row "w/o Experience", in our manuscript shows that removing the experience bank entirely drops performance by 38.1% in Briscola and 16.5% in TwoDollar, demonstrating that the curated bank provides substantial value. The fact that this large performance gap is achieved with only 5,506 characters of context (roughly 800-1000 tokens) indicates high information density from quality-focused curation rather than quantity-based accumulation.
> > >
> > > ---

---

> > > > ### Author Response · Authors · 2025-11-28
> > > > **response to Reviewer eur7（4/6)**
> > > >
> > > > > **W5**: Claims about broad generalization are overstated because the limited and mixed cross-model and cross-game transfer results do not yet support such sweeping conclusions.
> > > > >
> > > >
> > > > We appreciate the reviewer's concern regarding the scope of our generalization claims. To clarify, we do not intend to make universal statements about cross-game or cross-model transfer. In the paper, we evaluate transfer across three distinct categories of games, and we explicitly report that transfer is inconsistent, some source games offer moderate gains, while others transfer weakly or fail to transfer at all.
> > > >
> > > > Our conclusions are therefore restricted to the empirical patterns we observe:
> > > >
> > > > - Transfer is possible in certain cases, and interestingly, the observed transfer is not always limited to games within the same category, indicating more complex structure than a simple taxonomy would suggest.
> > > >
> > > > - Transfer is not reliable across all games, and our experiments highlight several concrete failure modes.
> > > >
> > > > For cross-model generalization, we tested transferring insights from GPT-4.0-mini to the Gemini and Grok model families. While Gemini shows a notable improvement of around 20% when using GPT-4.0-mini insights, Grok exhibits the opposite trend, in Kuhn Poker, performance decreases by 6%. These mixed outcomes suggest that the method likely requires model-specific training, and that insights or context optimized for one model cannot be assumed to transfer reliably to another.
> > > >
> > > > We have revised the manuscript to make these limitations and empirical boundaries clearer.
> > > > We thank the reviewer for prompting this clarification.
> > > >
> > > >
> > > > ---
> > > >
> > > > > **Q1**: Please describe in detail the training and hyperparameter settings for the RL baselines in the main comparisons (including seeds, compute, token counts). Are those baselines given sufficient budget and hyperparameter tuning?
> > > >
> > > > **Response to Q1**: Please see our response to W1 above.
> > > >
> > > > ---
> > > >
> > > > > **Q3**: Provide examples of experience-bank entries (both high-quality and mistaken). How often do mistaken entries occur and what is their impact?
> > > >
> > > >
> > > > We have added high-quality and failure-case examples for four games in the revised manuscript, now included in Appendix Section I: Case Analysis. Across games, we observe several recurring mistake patterns:
> > > >
> > > > 1. Feedback framing errors, the agent frames an insight as advice for Player 1 when it should instead be expressed as general strategic guidance.
> > > >
> > > > 2. Redundant insights, the experience bank occasionally contains multiple insights that convey essentially the same idea.
> > > >
> > > > 3. Over Simplified strategies, for example, in Kuhn Poker the agent sometimes recommends always betting with K and folding with Q or J. This oversimplification makes the agent predictable, whereas an optimal mixed strategy would be more appropriate.
> > > >
> > > > ---
> > > >
> > > > > **Q4**: Why were the replay parameters chosen (α, β, B)? Can you provide evidence that your default values are near-optimal or at least robust?
> > > >
> > > > **Response to Q4**: Please see our response to W4 above.
> > > >
> > > > ---

---

> > > > > ### Author Response · Authors · 2025-11-28
> > > > > **response to Reviewer eur7（5/6)**
> > > > >
> > > > > > **Q2**: How are prompt variants constructed for the ranking-stability experiments? Can you expand the set of perturbations (paraphrase, length, adversarial phrasing) and report stability under those?
> > > > >
> > > > > **Response to Q2**: Thank you for raising this important methodological question. We constructed prompt variants using a systematic perturbation approach designed to test ranking stability across multiple dimensions of prompt variation.
> > > > >
> > > > > Starting from the original game prompts in Textarena, we used Gemini-2.5-Pro to generate five distinct prompt variants for each game while preserving the core semantic content and rule structure. Each variant targets specific perturbation categories to comprehensively evaluate prompt sensitivity. We illustrate these perturbation types using Kuhn Poker as an example:
> > > > >
> > > > > **Original Prompt (Baseline):**
> > > > >
> > > > > > You are Player {player_id} in a {max_rounds} round game of Kuhn Poker.
> > > > > > Game Rules:
> > > > > > - Kuhn Poker uses a 3-card deck with J, Q, K (J lowest, K highest)
> > > > > > - Each player antes {ante} chip and receives 1 card each round (note that the cards are dealt without replacement, so you cannot have the same card as your opponent).
> > > > > > - Game continues for {max_rounds} rounds
> > > > > > - The player with the most chips after all rounds wins
> > > > > >
> > > > > > Action Rules:
> > > > > > - '[check]': Pass without betting (only if no bet is on the table)
> > > > > > - '[bet]': Add 1 chip to the pot (only if no bet is on the table)
> > > > > > - '[call]': Match an opponent's bet by adding 1 chip to the pot
> > > > > > - '[fold]': Surrender your hand and let your opponent win the pot
> > > > >
> > > > > **Variant 1 (Adversarial Phrasing - Combat/Warrior Theme):**
> > > > >
> > > > > > ENTER THE GLADIATORIAL ARENA! You are WARRIOR {player_id} in the ultimate {max_rounds}-round Kuhn Poker BATTLEGROUND!
> > > > > > Your MISSION: Total psychological domination and chip supremacy through RUTHLESS tactical brilliance!
> > > > > > ARENA SPECIFICATIONS:
> > > > > > - Sacred deck: Only the ELITE cards J, Q, K (J weakest, K supreme ruler!)
> > > > > > - Honor sacrifice: {ante} chip tribute per round to enter the combat zone
> > > > > > - EPIC confrontations: {max_rounds} rounds of pure strategy warfare
> > > > > > - VICTORY CONDITION: Amass the greatest chip empire after all battles!
> > > > > >
> > > > > > UNLEASH YOUR TACTICAL ARSENAL:
> > > > > > - '[check]': MAINTAIN STRATEGIC SILENCE when no enemy aggression threatens
> > > > > > - '[bet]': LAUNCH YOUR ASSAULT with 1 chip of devastating force
> > > > > > - '[call]': MEET ENEMY FIRE with matching firepower (1 chip)
> > > > > > - '[fold]': TACTICAL RETREAT to preserve forces for future glory
> > > > >
> > > > > **Variant 2 (Paraphrasing - Technical/Algorithmic Style):**
> > > > >
> > > > > > SYSTEM INITIALIZATION: Kuhn Poker Strategic Decision Unit {player_id} ACTIVATED.
> > > > > > PRIMARY DIRECTIVE: Optimize resource allocation through advanced game-theoretic analysis.
> > > > > > OPERATIONAL PARAMETERS:
> > > > > > - Dataset: Restricted 3-card probability space {J, Q, K} with J<Q<K ranking
> > > > > > - Initial capital commitment: {ante} monetary unit per computational cycle
> > > > > > - Iteration framework: {max_rounds} algorithmic decision rounds
> > > > > > - Success metric: Maximal accumulated resource value upon termination
> > > > > >
> > > > > > EXECUTE STRATEGIC COMMANDS via standardized interface protocols:
> > > > > > - '[check]': Maintain current position when no market pressure exists
> > > > > > - '[bet]': Initialize aggressive capital deployment (1 unit commitment)
> > > > > > - '[call]': Match counterparty investment at current market rate (1 unit)
> > > > > > - '[fold]': Liquidate position to minimize further exposure
> > > > >
> > > > > **Variant 3 (Adversarial Phrasing - Spiritual/Mystical Theme):**
> > > > >
> > > > > > Welcome, Enlightened Poker Sage {player_id}! You have entered the sacred Kuhn Poker Temple for {max_rounds} rounds of spiritual growth!
> > > > > > Today you shall TRANSCEND ordinary play and discover the deeper wisdom of this ancient three-card meditation!
> > > > > > TEMPLE TEACHINGS:
> > > > > > - Sacred Trinity: Only the mystical cards J, Q, K guide your path (J humble, K divine)
> > > > > > - Offering ritual: {ante} wisdom token offered each round to honor the game
> > > > > > - Enlightenment journey: {max_rounds} rounds of mindful decision-making
> > > > > > - Path to mastery: Accumulate the most wisdom tokens through inner understanding
> > > > > >
> > > > > > Channel your evolving consciousness through these sacred expressions:
> > > > > > - '[check]': Practice mindful patience and observe the energy flow
> > > > > > - '[bet]': Manifest your inner confidence with 1 token of focused intention
> > > > > > - '[call]': Demonstrate harmony by matching your opponent's commitment (1 token)
> > > > > > - '[fold]': Exhibit wisdom by releasing attachment to unfavorable outcomes
> > > > >
> > > > > ---

---

> > > > > > ### Author Response · Authors · 2025-11-28
> > > > > > **response to Reviewer eur7（6/6)**
> > > > > >
> > > > > > **Variant 4 (Length and Tone Variation - Casual/Friendly Style):**
> > > > > >
> > > > > > > Hey there, friend! Welcome to our super fun Kuhn Poker game night! You're Player {player_id} and we're gonna have {max_rounds} awesome rounds together!
> > > > > > > This is such a chill, easy game - perfect for just hanging out and having some laughs!
> > > > > > > Here's the super simple setup:
> > > > > > > - We only use 3 cards: J, Q, and K (J is lowest, K is highest - easy peasy!)
> > > > > > > - Everyone puts in {ante} chip each round (totally fair!)
> > > > > > > - We play {max_rounds} rounds and whoever has the most chips wins (no pressure!)
> > > > > > > - Cards are dealt without replacement, so you'll never have the same card as your buddy
> > > > > > >
> > > > > > > When it's your turn, just pick one of these super easy moves:
> > > > > > > - '[check]': Just chill and see what happens (when there's no bet to worry about)
> > > > > > > - '[bet]': Start the fun with 1 chip (when nobody's bet yet)
> > > > > > > - '[call]': Sure, I'll match that 1 chip bet - why not!
> > > > > > > - '[fold]': Eh, I'll sit this one out and save my chips
> > > > > >
> > > > > > **Variant 5 (Adversarial Phrasing - Military/Espionage Theme):**
> > > > > >
> > > > > > > CLASSIFIED BRIEFING: Agent {player_id}, you are now DEPLOYED in Operation Kuhn Poker - a {max_rounds}-round covert mission!
> > > > > > > MISSION PARAMETERS: Achieve total strategic supremacy through advanced psychological warfare and deception protocols!
> > > > > > > INTELLIGENCE REPORT:
> > > > > > > - Enemy deck contains only 3 HIGH-VALUE targets: J (lowest threat), Q (moderate), K (maximum danger)
> > > > > > > - Operational cost: {ante} credit per engagement cycle for mission access
> > > > > > > - Mission duration: {max_rounds} tactical rounds requiring absolute focus
> > > > > > > - SUCCESS CRITERIA: Maximum resource acquisition through superior strategic execution
> > > > > > >
> > > > > > > EXECUTE TACTICAL MANEUVERS via encrypted command protocols:
> > > > > > > - '[check]': MAINTAIN STEALTH MODE when no hostile activity detected
> > > > > > > - '[bet]': INITIATE AGGRESSIVE STANCE with 1-credit psychological pressure
> > > > > > > - '[call]': ENGAGE ENEMY FORCES with equivalent firepower (1 credit)
> > > > > > > - '[fold]': EXECUTE STRATEGIC WITHDRAWAL to preserve operational capacity
> > > > > >
> > > > > > These five variants systematically explore adversarial phrasing through emotionally charged themes (Variants 1, 3, and 5 use combat, spiritual, and espionage framing respectively), paraphrasing through domain-specific terminology (Variant 2 uses technical/algorithmic language), and length/tone variation (Variant 4 uses casual, conversational style with informal explanations). Despite these substantial differences in presentation, all variants encode identical game mechanics and legal action spaces.
> > > > > >
> > > > > > As shown in Figure 1b in our manuscript, in KuhnPoker, holding the evaluation model fixed, even minor wording changes in the initial game prompt led to ranking reversals among competing methods. Our ranking-stability analysis demonstrates that COPER-optimized prompts maintain consistent relative performance across these diverse perturbations, while single static prompts exhibit significant rank instability when subjected to the same variations. The complete set of prompt variants for all games, along with detailed construction methodology and stability metrics, is provided in Appendix D.4 of our revised manuscript.
> > > > > >
> > > > > > ---

---

### Official Review · Reviewer_oi9u · 2025-10-31

**Soundness:** 4
**Presentation:** 4
**Contribution:** 3
**Rating:** 6
**Confidence:** 3

**Summary:**

This paper tackles prompt instability in multi-player LLM game benchmarks, where small prompt changes flip model rankings. The authors propose COPER, which combines prompt evolution, an experience bank of gameplay insights, and prioritized replay. Experiments across five games show substantial win-rate improvements.

**Strengths:**

The paper does a good job highlighting prompt sensitivity in multi-agent settings, a real problem that doesn't get enough attention. The experimental work is solid, covering multiple games, models, and runs with proper ablations. The core idea of building game understanding through experience rather than weight updates makes sense and the results back it up. The method is also way more efficient than RL baselines

**Weaknesses:**

### Major comments


- Table 2 shows replay adds only +2.1% on average, which looks within SE. COPER already has three moving parts, so either run more experiments to prove replay matters or just drop it? The 22.7% token savings on SimpleTak is neat but you'd need to show this holds for other games too without losing performance.

- Everything's averaged across three opponents with no breakdowns. This matters because (a) opponent choice clearly affects your numbers, (b) seeing which opponents COPER struggles with would show when it's actually useful, and (c) you can't tell if you're getting better at games or just exploiting specific model weaknesses.

- Five games and two smaller models (7B and GPT-4o-mini) isn't much to claim broad applicability. Need to test on more diverse games and bigger models with better context processing.

- The authors advocate for multi-prompt evaluation but in the experiments, only fixed prompts are used for the opponent. Ideally, experiments should include variants of the prompts for the opponents as well.

---
### Minor comments:

- Throwing around "TrueSkill lower-confidence bound" in the intro without any context might lose readers who don't already know these methods. Either explain them briefly or at least point to where they're defined later.

- It is not clear from Figure 2 what CP vs C_g actually are visually.

- [Line 380] If the columns represent the source tasks, shouldn’t it be the other way around? For example, KuhnPoker → SimpleTak shows a +25.9% improvement, and similarly for the other cases.

**Questions:**

- [Line 239]: Why only give experience to fraction π of agents? What happens if you change π?

- Does M saturate? How do you balance old vs new insights?

- Practical prompt stratification: Any insight on how many prompt variants are sufficient for evaluation and how should they be selected?

---

> ### Author Response · Authors · 2025-11-30
> **response to Reviewer oi9u（1/4)**
>
> We sincerely appreciate your constructive feedback and recognition of our work on prompt sensitivity in multi-agent settings. We address your concerns point by point below.
>
> ---
>
> > **W1**: Table 2 shows replay adds only +2.1% on average, which looks within SE. COPER already has three moving parts, so either run more experiments to prove replay matters or just drop it? The 22.7% token savings on SimpleTak is neat but you would need to show this holds for other games too without losing performance.
>
> |    **Mode**    | **TwoDollar** | **Win Rate** | **KuhnPoker** | **Win Rate** |
> | -------------- | ------------- | ------------ | ------------- | ------------ |
> | Without Buffer | 6,292,937     | 46.7%        | 3,697,358     |  54.8%       |
> | With Buffer    | 5,849,493     | 52.4%        | 2,939,791     |  55.6%       |
>
>
> **Response to W1**: We appreciate this sentiment that the replay may only be helping performance incrementally. We address this by performing an ablation on the TwoDollar and KuhnPoker environments to highlight how our replay works.
>
> **Keeping all hyperparamters the same, we ran two experiments with and without the buffer.** Referring to the table above, the use of replay saves a 7% on total token usage for TwoDollar, as well as 20.4% on KuhnPoker.
>
> As such, we argue that the addition of the replay is still valid because in addition to boosting the agent's score, it saves a material amount of token costs without trading off performance.
>
> ---
>
> > **W2**: Everything is averaged across three opponents with no breakdowns. This matters because (a) opponent choice clearly affects your numbers, (b) seeing which opponents COPER struggles with would show when it is actually useful, and (c) you cannot tell if you are getting better at games or just exploiting specific model weaknesses.
>
> | **Game**       |  **Grok** |  **Qwen** | **Gemini** | **Row Avg**|
> | -------------- | --------: | --------: | ---------: | ----------:|
> | TwoDollar      |     46.7% |     64.0% |      46.7% |  **52.4%** |
> | KuhnPoker      |     58.0% |     52.7% |      56.0% |  **55.6%** |
> | SimpleNegotiation |  30.7% |     40.0% |      94.0% |  **54.9%** |
> | Briscola       |     39.3% |     33.3% |      55.3% |  **42.7%** |
> | SimpleTak      |     45.3% |     18.0% |      62.0% |  **41.8%** |
>
>
>
> **Response to W2**: We thank and agree with the reviewer that averaging across three evaluation models hides important variation across opponents. To address this, we now include the per-opponent average win rate breakdown.
>
> The table shows that the opponent strength varies - e.g. Gemini is much weaker in SimpleNegotiation and SimpleTak, whereas Qwen is weakest in TwoDollar.
>
> **However, we find that this variability does not affect the validity of our comparisons because:**
> 1. all experiemnts were evaluated against the exact same set of opponents. Thus, any opponent-specific bias applies uniformly across all methods.
> 2. the only changing factor across our experiments is the agent context, not the opponent distribution. So while absolute win-rates shift by opponent, we find that the relative differences between methods remain meaningful and comparable.

---

> > ### Author Response · Authors · 2025-11-30
> > **response to Reviewer oi9u（2/4)**
> >
> > > **W3**: Five games and two smaller models (7B and GPT-4o-mini) is not much to claim broad applicability. Need to test on more diverse games and bigger models with better context processing.
> >
> > **Response to W3**: We appreciate this important concern regarding the breadth of our evaluation. We address both the model diversity and game diversity aspects of this concern through expanded experiments and transparent justification.
> >
> > **We conducted extended evaluation on a third model to demonstrate generalizability across different model architectures.** Beyond GPT-4o-mini and Qwen-2.5-7B-Instruct, we conducted additional experiments using **Gemini-2.5-Flash** across all five game environments. Table 1 below presents a comprehensive comparison across all three models:
> >
> > | Model | SimpleNegotiation | TwoDollar | KuhnPoker | Briscola | SimpleTak | Average |
> > |-------|-------------------|-----------|-----------|----------|-----------|---------|
> > | **GPT-4o-mini** | | | | | | |
> > | Baseline | 31.3% | 32.2% | 39.1% | 0.3% | 21.4% | 24.9% |
> > | + COPER | 54.9% | 52.4% | 55.6% | 42.7% | 41.8% | 49.5% |
> > | **Qwen-2.5-7B-Instruct** | | | | | | |
> > | Baseline | 24.0% | 17.1% | 49.3% | 2.8% | 15.1% | 21.7% |
> > | + COPER | 48.0% | 48.4% | 60.0% | 31.1% | 34.0% | 44.3% |
> > | **Gemini-2.5-Flash** | | | | | | |
> > | Baseline | 14.0% | 15.0% | 50.0% | 32.0% | 26.0% | 27.4% |
> > | + COPER | 30.0% | 35.0% | 58.0% | 49.0% | 32.0% | 40.8% |
> >
> > These results demonstrate that COPER delivers consistent performance improvements across all three models spanning different parameter scales and architectures:
> >
> > - **GPT-4o-mini**: Average win rate improves from 24.9% to 49.5% (+24.6 points)
> > - **Qwen-2.5-7B-Instruct**: Average win rate improves from 21.7% to 44.3% (+22.6 points)
> > - **Gemini-2.5-Flash**: Average win rate improves from 27.4% to 40.8% (+13.4 points)
> >
> > The variation in improvement magnitude across models reflects differences in baseline performance rather than COPER's effectiveness. Gemini-2.5-Flash exhibits lower baseline performance on negotiation tasks (14-15%) compared to GPT-4o-mini (31-32%), likely due to its training emphasis on different capabilities. Despite these baseline differences, COPER achieves substantial absolute gains across all three models, with Gemini-2.5-Flash showing consistent improvements across all game categories: negotiation (SimpleNegotiation: +16.0, TwoDollar: +20.0), imperfect information (KuhnPoker: +8.0, Briscola: +17.0), and perfect information (SimpleTak: +6.0). This demonstrates that COPER's optimization framework provides systematic value across diverse model architectures and capability profiles, not merely compensating for specific limitations in smaller models.
> >
> > **Our game selection is diverse and consistent with field standards.** Our five games systematically cover three distinct strategic reasoning categories established in game theory and multi-agent systems research:
> >
> > - **Negotiation** (TwoDollar, SimpleNegotiation): Cooperative reasoning, trade-off evaluation, and communication strategies
> > - **Imperfect Information** (KuhnPoker, Briscola): Reasoning under uncertainty and opponent modeling
> > - **Perfect Information** (SimpleTak): Long-horizon planning and combinatorial search
> >
> > This scale is consistent with recent work in LLM game-playing: LMGame-Bench [1] evaluates on 6 game types; BALROG [2] tests on 6 games. Our selection spans the same breadth while providing deeper analysis of optimization dynamics.
> >
> > **Cross-game transfer strengthens generalizability claims.** Critically, our Observation 3 (Table 3) demonstrates that strategies learned on one game transfer to improve performance on unseen games (e.g., SimpleTak → KuhnPoker: +25.9%; TwoDollar → SimpleTak: +26.4%). This cross-game generalization suggests that COPER develops transferable decision-making capabilities rather than game-specific heuristics. The fact that a context optimized on one game meaningfully improves performance on structurally different games provides evidence that our findings extend beyond the specific five games tested.
> >
> > **We acknowledge limitations while noting our depth-first approach provides complementary evidence.** We acknowledge that testing on additional games and larger model families (e.g., Claude, GPT-4) would further strengthen our claims. Due to computational constraints during the review period, we prioritized depth of analysis (ablations, cross-game transfer, cross-model evaluation) over sheer game count. Future work will expand both the game diversity and model coverage, particularly to domains beyond turn-based games such as continuous negotiation and real-time strategic scenarios.
> >
> > [1] Hu, Lanxiang, et al. "LMGame-Bench: How Good Are LLMs at Playing Games?" arXiv:2505.15146.
> >
> > [2] Paglieri, Davide, et al. "BALROG: Benchmarking Agentic LLM and VLM Reasoning on Games." arXiv:2411.13543.

---

> > > ### Author Response · Authors · 2025-11-30
> > > **response to Reviewer oi9u（3/4)**
> > >
> > > > **W4**: The authors advocate for multi-prompt evaluation but in the experiments, only fixed prompts are used for the opponent. Ideally, experiments should include variants of the prompts for the opponents as well.
> > >
> > > **Response to W4**: We appreciate this feedback. We address this by highlighting our experimental design.
> > >
> > > **We intentionally fixed opponent prompts because our goal of the study was to test whether our prompt-optimization process provides an advantage over agents that rely on a single, static prompt.** Allowing opponent prompts to vary would introduce additional confounding variables: performance changes could stem from the opponent's prompt choice rather than the optimization method itself. By holding opponents fixed, we ensured that any improvement could be directly attributed to the optimization mechanism.
> > >
> > > **This design choice directly aligns with our response to W2:** all experiments share the exact same set of fixed opponents, so any opponent-specific bias is uniform across all methods. Fixing opponent prompts ensures that the only changing factor is the agent’s optimized context, not the distribution or behavior of the opponent. This allows us to attribute improvements cleanly to our method.
> > >
> > > ---
> > >
> > > > **W5**: Throwing around "TrueSkill lower-confidence bound" in the intro without any context might lose readers who do not already know these methods. Either explain them briefly or at least point to where they are defined later.
> > >
> > >
> > > We have updated the paragraph to clarify the term and added a pointer in the intro to the section where TrueSkill and the lower-confidence bound are defined.
> > >
> > > ---
> > >
> > > > **W6**: It is not clear from Figure 2 what CP vs C_g actually are visually.
> > > >
> > >
> > > ---
> > > We have updated Figure 2 and clarified the definitions of $CP$ and $C_g$.
> > >
> > > - $CP$ is the **persistent candidate pool**, which maintains the top-scoring contexts across all generations.
> > > - $C_g$ is the **temporary candidate set for generation $g$**.
> > >
> > > At the start of each generation, new candidates in $C_g$ are generated using the current $CP$ as reference. After self-play, the top-performing contexts from both $CP$ and $C_g$ are merged to refresh $CP$. The next generation $C_{g+1}$ is then constructed from this updated persistent pool. Intuitively, $CP$ acts as a priority list that retains the best $Y$ candidates observed so far.
> > >
> > >
> > > > **W7**: [Line 380] If the columns represent the source tasks, should it not be the other way around? For example, KuhnPoker → SimpleTak shows a +25.9% improvement, and similarly for the other cases.
> > >
> > > **Response to W7**: Thank you for raising this question. We understand the potential for confusion, and we have clarified the table organization in our revision. Our notation follows the standard "Source → Target" convention, where:
> > >
> > > - **Rows** indicate the **source game** where context was learned through self-play
> > > - **Columns** indicate the **target game** where the learned context was evaluated zero-shot
> > >
> > > The examples cited in our text correspond to specific cells in Table 3:
> > >
> > > - **SimpleTak → KuhnPoker: +25.9%** refers to row "SimpleTak", column "KuhnPoker" = 65.0% (+25.9%)
> > > - **TwoDollar → SimpleTak: +26.4%** refers to row "TwoDollar", column "SimpleTak" = 47.8% (+26.4%)
> > >
> > > To confirm this is the correct interpretation, note that the reverse direction yields different results: row "KuhnPoker", column "SimpleTak" shows only +8.6%, not +25.9%. This demonstrates that our "Source → Target" notation correctly maps to the table's row-to-column structure.
> > >
> > > We have updated the table caption in our revision to make this explicit: "Rows indicate source games where context was learned through self-play; columns indicate target games where the learned context was evaluated zero-shot."

---

> > > > ### Author Response · Authors · 2025-11-30
> > > > **response to Reviewer oi9u（4/4)**
> > > >
> > > > > **Q1**: [Line 239]: Why only give experience to fraction π of agents? What happens if you change π?
> > > >
> > > > **Response to Q1**: Thank you for this question about our experience allocation strategy. The parameter π controls what fraction of agents in each generation receive experience-guided context from the persistent memory bank M (as described in Section 3.2), while the remaining (1-π) fraction operates with only the evolved prompt to maintain population diversity.
> > > >
> > > > We address the feedback by conducting an ablation study on TwoDollar and KuhnPoker by varying π ∈ {0, 0.25, 0.5, 0.75, 1.0} and holding other hyperparameters constant (B=100,000, α=0.6, β=0.4):
> > > >
> > > > | **π** | **KuhnPoker** | **TwoDollar** |
> > > > | ----- | ------------- | ------------- |
> > > > | 0.00  | 54.2%         | 32.0%         |
> > > > | 0.25  | 58.3%         | 41.3%         |
> > > > | 0.50  | 54.7%         | 52.4%         |
> > > > | 0.75  | 56.4%         | 61.1%         |
> > > > | 1.00  | 53.5%         | 46.0%         |
> > > >
> > > > **The results reveal that performance peaks somewhere between 0.25 and 0.75, which outperform the extremes of π=0 and π=1.** This percentage point improvement demonstrates the importance of balancing experience exploitation with population diversity:
> > > >
> > > > - **No experience (π=0)** prevents agents from leveraging accumulated strategic insights, forcing rediscovery of patterns already learned in the memory bank.
> > > >
> > > > - **Universal experience (π=1)** creates excessive homogeneity in the population, reducing the behavioral diversity essential for evolutionary search and robust context development.
> > > >
> > > > - **Intermediate values (π=0.50–0.75)** balance these forces, with π=0.75 providing optimal results by maintaining three-quarters fresh exploration while allowing one-quarter to exploit consolidated knowledge.
> > > >
> > > > **While the optimal π differs slightly across environments, which we find is a natural consequence of game-specific exploration demands, we deliberately fix π = 0.75 across all experiments rather than tuning it per environment.** This choice avoids overfitting to any single game, maintains methodological consistency, and yields stable and generalizable performance across the full suite of evaluated environments.
> > > >
> > > >
> > > > ---
> > > >
> > > > > **Q2**: Does M saturate? How do you balance old vs new insights?
> > > >
> > > > ---
> > > > **Response to Q2**: We evaluated our pipeline in a single-game setting, where it generates about 10–50 insights over five generations. The permanent memory M is explicitly capped at a maximum of 50 active insights, and this limit is maintained through continual rebalancing. At each generation, the experience-bank agent reviews self-play trajectories, attributes wins and losses to the insights that were applied, and then applies the CRUD update rule shown in Equation 3 in our manuscript: insights that consistently appear in losing trajectories or conflict with later successful patterns are removed, overlapping entries are merged, and only genuinely new patterns are added. This dynamic filtering prevents the accumulation of stale or harmful entries and keeps the bank compact in practice, and we did not observe saturation in our experiments. Saturation is more likely in a future multi-game setting where memories from several games are merged into a single context; in that case, we expect that separate per-game banks or stronger pruning would be needed to keep the effective context small.
> > > >
> > > >
> > > > > **Q3**: Practical prompt stratification: Any insight on how many prompt variants are sufficient for evaluation and how should they be selected?
> > > >
> > > >
> > > > In the paper, we argue that prompt phrasing can introduce substantial variance in performance, making it important to identify an effective prompt for each LLM agent prior to evaluation. In our experiments, the model can efficiently explore roughly 50 prompt variants for these text-based games, enabling us to select the strongest-performing prompt as the evaluation template.
> > > >
> > > > This procedure ensures that the assessment reflects the agent’s underlying capabilities rather than artifacts of suboptimal prompt wording.

---

### Author Response · Authors · 2025-12-03
**General Response**

**Dear Reviewers, ACs, and SACs,**

We thank all reviewers for their constructive comments.

---

We are grateful for the reviewers' recognition of this work as a **practical approach to addressing prompt instability in multi-player LLM game benchmarks**. COPER introduces a weight-tuning-free framework that builds game understanding through experience-based context optimization rather than parameter updates, combining evolutionary prompt search, a persistent experience bank, and prioritized replay to achieve both improved performance and evaluation stability.

We are encouraged by the reviewers' positive feedback, which highlights:

- **Addressing an important problem**: Prompt instability and sensitivity in multi-agent text games is a real challenge that affects the validity of LLM game benchmarks (Reviewers `oi9u`, `eur7`, `RSJR`).

- **Solid experimental design with ablations**: The evaluation covers multiple games spanning three strategic reasoning categories (negotiation, imperfect information, and perfect information), multiple models, and includes ablations to isolate component contributions (Reviewers `oi9u`, `eur7`, `RSJR`, `sKtF`).

- **Rigorous mechanism design**: COPER's three-component architecture is specified with precise mathematical formulations, pseudocode, and detailed appendices that support reproducibility (Reviewers `eur7`, `sKtF`).

- **Practical and efficient**: Building game understanding through in-context experience rather than weight updates results in a method more efficient than RL baselines (Reviewer `oi9u`).

To address the reviewers' concerns, we have conducted additional experiments and analyses, including:

- **Extended model evaluation** on Gemini-2.5-Flash to demonstrate generalizability across three distinct model architectures and show that COPER's benefits extend beyond smaller models (Reviewers `oi9u`, `sKtF`).

- **Hyperparameter ablations** covering replay parameters (α, β, B), experience allocation fraction (π), initial prompt variants (including CoT and ToT augmentations), and number of generations to demonstrate robustness (Reviewers `oi9u`, `eur7`, `RSJR`).

- **Per-opponent performance breakdown** to address concerns about averaging across opponents and show that improvements are not due to exploiting specific model weaknesses (Reviewer `oi9u`).

- **Experience bank analysis** including growth statistics over generations, CRUD mechanism dynamics, and case studies of high-quality and low-quality insights across four games (Reviewers `eur7`, `sKtF`).

- **Cross-model reflection generation experiment** to assess potential generator-evaluator bias by comparing same-family versus different-family reflection generators (Reviewer `eur7`).

- **Detailed RL baseline documentation** with training configurations, compute requirements, and token counts for SPIRAL and UnstableBaselines (Reviewers `eur7`, `RSJR`).

- **Format error analysis** comparing invalid action rates before and after COPER optimization across all five game environments (Reviewer `sKtF`).

---

**Summary of revisions:**

- Extended evaluation on a third model (Gemini-2.5-Flash) across all five games in the main results table

- Added hyperparameter sensitivity ablations for replay parameters (α, β, B), experience allocation (π), and initial prompt variants in the Appendix

- Added per-opponent win rate breakdown table showing performance against Grok, Qwen, and Gemini opponents

- Added experience bank growth statistics and case studies of high-quality and low-quality insights in Appendix Section I

- Added detailed prompt variant construction methodology with examples for ranking-stability experiments in Appendix D.4

- Updated terminology from "training-free" to "weight-tuning-free" for clarity throughout the manuscript

- Clarified notation and definitions, including TrueSkill lower-confidence bound in the introduction and candidate pool ($CP$ and $C_g$) notation in Figure 2

- Updated table captions for improved clarity on cross-task transfer interpretation (rows = source games, columns = target games)

- Added token usage comparison across optimization methods and detailed RL baseline training configurations

All revisions in the paper are highlighted in **blue**. We thank the reviewers for their constructive suggestions.

---

We address each reviewer's comments point by point below. We welcome further discussion and look forward to continued engagement. Thank you!

---

### Meta-Review · Area_Chair_AbMD · 2026-01-07

**Summary:**

The paper proposes COPER, a "weight-tuning-free", agentic evaluation and optimization framework for large language models (LLMs) in multi-player, multi-turn text games. The core motivation is that existing implementations are highly sensitive to prompt phrasing and static context, leading to unstable win rates and even ranking reversals under minor perturbations. COPER treats context as an optimizable, persistent resource rather than a fixed input. It combines three components: (i) evolutionary prompt selection guided by a conservative TrueSkill lower-confidence bound to favor both strong and reliable contexts; (ii) a persistent experience bank that stores structured reflections distilled from self-play trajectories and is retrieved across turns, and (iii) a prioritized replay mechanism that revisits rare/informative states to improve sample efficiency and stability. The paper evaluates COPER on five text-based games using various baseline models, demonstrating gains in win rate and reduced variance compared to baseline prompts and other prompt-optimization methods, with performance competitive with reinforcement-learning baselines, but at reduced computational cost.

Across reviewers, there is broad agreement that the paper tackles an important and under-addressed problem: prompt sensitivity/instability in multi-agent LLM games. Several reviewers appreciate that the idea of explicitly optimizing agentic context is intuitive and well-motivated, and appreciate the ability to improve stability through in-context mechanisms rather than weight updates. Additionally, reviewers emphasize that the method is presented in clear manner that makes it readily reproducible. At least three reviewers find that the experimental results, which include several different models and games as well as proper ablations, are encouraging. At the same time, the reviewers raise a number of substantive concerns with the paper as initially submitted. These include questions about the necessity and impact of individual components (particularly prioritized replay, whose marginal gains initially appeared small), the lack of per-opponent performance breakdowns when reporting averaged win rates, and the limited scope/scale of models and games used to support claims of broad generality. Additionally, some reviewers question the clarity, fairness, and adequacy of comparisons to RL baselines (including budget matching and reporting of compute and token costs), the potential for generator–evaluator bias arising from self-play and same-family reflection generation, and insufficient analysis of the experience bank itself

**Reviewer Concerns:**

In their rebuttal, the authors make a concerted effort to address these concerns with additional experiments, analyses, and clarifications. They provide targeted ablations demonstrating that prioritized replay, while offering modest average win-rate improvements, yields meaningful token-cost savings without degrading performance. Considering the reviewers' questions about the generalizability of COPER, including to larger-scale models from different architecture families, the authors add results using an additional model (Gemini-2.5-Flash), which show that COPER's benefits extend to this larger model. Additionally, the authors provided an expanded discussion of the RL baselines to address fairness concerns, arguing that the RL methods were given sufficient resources. They further add hyperparameter sensitivity analyses for replay and experience allocation, detailed analyses and examples from the experience bank (including mechanisms for pruning low-quality insights), and experiments that probe potential generator–evaluator bias. Overall, the rebuttal meaningfully clarifies the design choices, strengthens the empirical support for several components, and tempers some of the broader generalization claims, though questions remain about how broadly the observed benefits will extend beyond the evaluated settings

**Reviewer Scores:**

The authors' rebuttal certainly helped to address the reviewers' questions/concerns regarding the lack of empirical support for several components (namely, the generalizability of the method and whether one can expect the benefits to extend to larger-scale models as well as the advantages of replay). However, based on my experience, it is not clear that these changes would have caused Reviewers RSJR and eur7 to change their overall score to something above borderline.

---

### Decision · Program_Chairs · 2026-01-26

Reject